# Assessment of a tiling energy budget approach in a land surface model, ORCHIDEE-MICT (r8205)

Yi Xi[1], Chunjing Qiu[2,3], Yuan Zhang[4,1], Dan Zhu[5], Shushi Peng[5], Gustaf Hugelius[6], Jinfeng Chang[7], Elodie Salmon[1], Philippe Ciais[1]

[1]Laboratoire des Sciences du Climat et de l'Environnement, LSCE/IPSL, CEA – CNRS – UVSQ, Université Paris-Saclay, 91191 Gif-sur-Yvette, France
[2]Research Center for Global Change and Complex Ecosystems, School of Ecological and Environmental Sciences, East China Normal University, Shanghai, China.
[3]Institute of Eco-Chongming, East China Normal University, Shanghai, China
[4]Key Laboratory of Alpine Ecology, Institute of Tibetan Plateau Research, Chinese Academy of Sciences, Beijing 100101, China;
[5]Sino-French Institute for Earth System Science, College of Urban and Environmental Sciences, Peking University, Beijing 100871, China
[6]Department of Physical Geography, Stockholm University, Stockholm, Sweden
[7]College of Environmental and Resource Sciences, Zhejiang University, Hangzhou, China

Correspondence to: Yi Xi (yi.xi@lsce.ipsl.fr)

**Abstract.** The surface energy budget plays a critical role in terrestrial hydrologic and biogeochemical cycles. Nevertheless, its highly spatial heterogeneity across different vegetation types is still missing in the land surface model, ORCHIDEE-MICT (ORganizing Carbon and Hydrology in Dynamic EcosystEms–aMeliorated Interactions between Carbon and Temperature). In this study, we describe the representation of a tiling energy budget in ORCHIDEE-MICT, and assess its short and long-term impacts on energy, hydrology, and carbon processes. With the specific values of surface properties for each vegetation type, the new version presents warmer surface and soil temperatures (~0.5 °C, 3%), wetter soil moisture (~10 kg m$^{-2}$, 2%), and increased soil organic carbon storage (~170 PgC, 9%) across the Northern Hemisphere. Despite reproducing the absolute values and spatial gradients of surface and soil temperatures from satellite and in-situ observations, the considerable uncertainties in simulated soil organic carbon and hydrologic processes prevent an obvious improvement of temperature bias existing in the original ORCHIDEE-MICT. However, the separation of sub-grid energy budgets in the new version improves permafrost simulation greatly by accounting for the presence of discontinuous permafrost types (~ 3 million km$^2$), which will facilitate various permafrost-related studies in the future.

# 1 Introduction

The surface energy balance is a fundamental component of the Earth system. The incoming solar energy is not only essential for life and plant photosynthesis but also drives the terrestrial hydrologic cycle (i.e., evapotranspiration and the freeze-thaw cycle of soils in cold regions) and modulates the speed of biogeochemical reactions (i.e., the decomposition of organic matter). The energy balance depends on landscape type through distinct vegetation and soil elements which reflect and emit shortwave and longwave electromagnetic radiation in different proportions. Understanding and simulating the complex interactions of energy, hydrology, and biogeochemical processes throughout the Earth system is crucial for tracking the consequences of historical human activities and predicting the future of our Earth's climate system.

Employing land surface models (LSMs) or earth system models (ESMs) is one of the most common approaches to simulate the surface energy budget and investigate its interactions with hydrologic, atmospheric and biogeochemical processes. The typical spatial resolution of the LSMs varies from 0.5º × 0.5º (~50 km × 50 km at equator) to 2º × 2º (~200 km × 200 km at equator). Significant surface heterogeneity would undoubtedly exist on such large scales. Taking surface temperature ($T_{surf}$) as an example, in reality, two adjacent landscapes could have significantly different $T_{surf}$ due to their distinct surface properties, including surface albedo, leaf area index, rooting depth, vegetation height at scale not resolved by the models. For instance, the larger latent heat loss via evapotranspiration over deep rooted tropical forests compared to nearby grassland and cropland shows a significant cooling effect, approximately 2.5 °C on a daily basis (Li et al., 2015). The higher albedo across snow-covered areas for short vegetation compared to nearby forest results in a reduction in the absorbed solar energy and lower the $T_{surf}$, with a magnitude depending on the timing and duration of snow cover (Zhang, 2005). To represent the heterogeneous surface energy balance, some LSMs / ESMs have introduced tiling energy budgets such as the PFT-specific energy budgets in CLASSIC (Canadian Land Surface Scheme Including Biogeochemical Cycles) (Melton and Arora, 2014), the separate energy budgets for snow, soil, and vegetation in ISBA (Soil–Biosphere–Atmosphere LSM) (Boone et al., 2017), the partition of snow-cover and snow-free land units in CLM 5.0 (Community Land Model) (Lawrence et al., 2019), and the sub-grid topographic effects on solar radiation flux in ELM (Energy Exascale Earth System Model (E3SM) Land Model) (Hao et al., 2021). Moreover, three land surface schemes (LSSs) have been adopted to represent the tiling energy budgets including mosaic (use specific surface properties for each land cover type), mixed (grouping certain land cover types with similar surface properties and then having a smaller number of distinct surface types), and composite (using the average properties of one grid cell) (Melton and Arora, 2014; Rumbold et al., 2023). Through the comparison between the "mosaic" and the "composite" LSSs, the CLASSIC model reported a less than 5% difference in the primary energy fluxes but an up to 46% difference in carbon fluxes and carbon pool size at site level (Li and Arora, 2012), as well as a 19% higher terrestrial carbon sink for 1959-2005 in the "mosaic" simulation (Melton and Arora, 2014). Rumbold et al. (2023) also found that the tiling soil scheme does have an impact on the water and energy budgets due to the way vegetation accesses soil moisture with the JULES model (Joint UK Land Environment Simulator). Besides, Qin et al. (2023) found that the tiling CLM model provides more accurate simulations

of surface air temperature and precipitation than the single-land-cover version when coupled with the WRF model (Weather Research and Forecasting), as validated against in-situ observations. Despite uncertainties in model-specific structures and configurations, these findings highlight the importance of representing explicitly sub-grid surface heterogeneity in current LSMs.

Besides the necessity of representing surface heterogeneity, the incorporation of new landforms and processes also requires the tiling of energy budgets. For instance, Rooney and Jones (2010) identified the challenges in simulating soil temperature under lakes when introducing the lakes into the single-soil-tile JULES, since they have different thermal transfer characteristics due to the higher specific heat capacity of water than adjacent land tiles. When evaluating the impacts of subgrid-scale disturbances such as fires and harvest, Curasi et al. (2023) found the impact of sub-grid heterogeneity is 1.5 to 4 times the
impact of disturbances themselves on the carbon cycle with the CLASSIC model. Besides, it's necessary to provide the independent energy budgets, hydrology and carbon cycles when incorporating a series of new processes for permafrost regions such as discontinuous permafrost type (Smith et al., 2022), melting of ground ice (Rumbold et al., 2023), thermokarst thawing and lateral drainage (Nitzbon et al., 2020) in LSMs.

To enable the representation of surface heterogeneity and open the door to a series of new landforms and processes, we implement tiling energy budgets at the surface and subsurface for each plant function type (PFT) in a state-of-the-art LSM, ORCHIDEE-MICT (ORganizing Carbon and Hydrology in Dynamic EcosystEms–aMeliorated Interactions between Carbon and Temperature), which calculates turbulent fluxes for subgrid PFT types but solves for an average grid-level energy budget resulting into a single surface temperature (Guimberteau et al., 2018). Some hydrologic and biogeochemical processes in the
model have been modified correspondingly to include PFT-specific thermal inputs. In Sect. 2, we provide a brief review of the current grid-cell mean energy budget at surface and subsurface, while Sect. 3 describes the modifications made to implement the tiling energy budgets in the model. To evaluate the impacts of separating the energy budget for each PFT on the energy, hydrology, soil thermics and carbon cycles, we conduct simulations using the original version of ORCHIDEE-MICT (referred to as MICT) and the new tiling energy budget version (referred to as MICT-teb) as described in Sect. 4. The results of these
simulations are compared in Sect. 5. Section 6 focuses on the evaluation of energy processes in the new version as well as the improvements for permafrost simulations. Finally, we present conclusions in Sect. 7.

## 2 Overview of current energy budget in ORCHIDEE-MICT

ORCHIDEE-MICT is a branch of the ORCHIDEE model (Krinner et al., 2005) specifically developed to enhance the
95 representation of hydrologic and biogeochemical interactions in high latitude regions (Guimberteau et al., 2018). In comparison to the trunk version 3976 from which it was developed, ORCHIDEE-MICT includes several key new processes, namely 1) the

feedback effects of soil organic carbon (SOC) concentration on soil thermal and soil water dynamics (Zhu et al., 2019); 2) soil carbon vertical discretization (Koven et al., 2009; Zhu et al., 2016); 3) vertical mixing of soil carbon due to cryoturbation (in cold soils) and bioturbation (Koven et al., 2009, 2013); 4) reformulation of soil hydric stress above the permafrost table (Zhu et al., 2015); 5) the inclusion of northern peatlands as a specific PFT with peatland-specific hydrology, carbon decomposition and accumulation (Qiu et al., 2018, 2019). These new processes have significantly improved the representation of plant productivity, water cycle, soil carbon stocks, and the simulated permafrost distribution in high latitude regions, but there is still room for improvement (Guimberteau et al., 2018). One important aspect that calls for attention is the need to include sub-grid representations of surface and subsurface energy budgets, especially in the high latitudes where snow cover and water equivalent differs between PFTs, and where soil carbon differences across landscape elements / PFTs within the same grid-cell results into heterogeneous soil temperature and active layer thickness. For the moment, within the ORCHIDEE-MICT model, there are three major modules: carbon, water, and energy. The carbon cycle module operates for each PFT sub-grid and the water cycle operates for each soil tile (divided into bare soil, tree PFTs, grass and crop PFTs, and peatland PFT categories), while the energy module is solved only at the total grid-cell level (Best et al., 2004). Figure 1 shows the schematic representation of energy budgets from the surface to snow and soil layers in the original and new versions of ORCHIDEE-MICT, namely MICT and MICT-teb. The details of the energy budget in the model are described as follows.

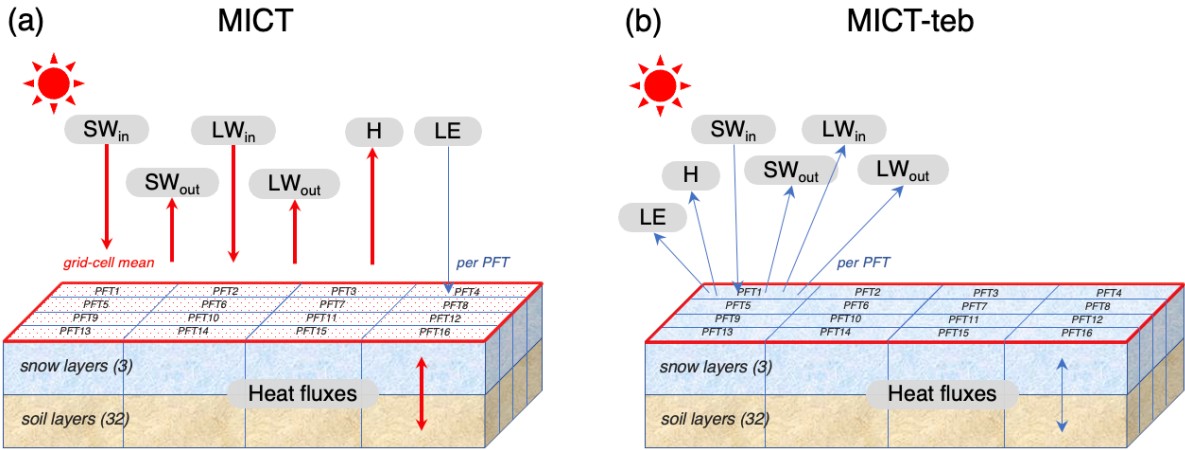

**Figure 1. Schematic representation of energy budgets at the surface, snow layers, and soil layers in one grid cell of ORCHIDEE-MICT (MICT) (a) and the new tiling energy budget version (MICT-teb) (b). SW$_{in}$, SW$_{out}$, LW$_{in}$, LW$_{out}$, H, and, LE represent incoming ShortWave radiation, outward ShortWave radiation, incoming LongWave radiation, outward LongWave radiation, sensible heat flux, latent heat flux, respectively. PFT indicates Plant Function Type. There are 3 layers for snow, and 32 layers for soil for each PFT in the model. In MICT, SW$_{in}$, SW$_{out}$, LW$_{in}$, LW$_{out}$, H, and heat fluxes in snow and soil layers are calculated as grid-cell mean but LE is calculated for each PFT, while in MICT-teb, all of the heat fluxes are calculated for each PFT. The red and blue arrows distinguish the grid-cell mean and PFT-specific calculation.**

## 2.1 Surface energy budgets

Similar to most LSMs, the surface energy budget equation is used in MICT to describe the balance of net absorbed radiation ($R_{net}$) by the energy transferred out of the ecosystem:

$$R_{net} = H + LE + G + \Delta S \tag{1}$$

where H is sensible heat flux (W m$^{-2}$); LE is latent heat flux (W m$^{-2}$); G is ground heat flux (W m$^{-2}$); $\Delta S$ is the energy stored in the ecosystem as chemical energy through photosynthesis and as the temperature change of the plant biomass (W m$^{-2}$). Due to the $\Delta S$ only accounting for < 5% of total energy budget of the ecosystem in most areas (Georg et al., 2016), it is neglected in the model. $R_{net}$ is the balance of the inputs and outputs of shortwave radiation ($SW_{in}$ and $SW_{out}$) and longwave radiation ($LW_{in}$ and $LW_{out}$):

$$R_{net} = (SW_{in} - SW_{out}) + (LW_{in} - LW_{out}) = (1 - \alpha) \times SW_{in} + (LW_{in} - \sigma \times \varepsilon \times (T_{surf})^4) \tag{2}$$

where $\alpha$ is the albedo which determines the proportion of incoming SW absorbed by the ecosystem (unitless); $\sigma$ is Stefan-Boltzman constant (5.6697×10$^{-8}$ W m$^{-2}$ K$^{-4}$); $\varepsilon$ is emissivity (1, unitless) and $T_{surf}$ is the surface temperature (K). In MICT, the grid-cell $\alpha$ is calculated as the area-weighted average of the $\alpha$ across all PFTs. The snow-covered areas and snow-free areas are distinguished in terms of the higher albedo of snow than canopy and bare soil (see details about the calculation of albedo for snow-covered regions in Sect. 2.3). The albedo of bare soil in the current version is prescribed using static satellite observations (Lurton et al., 2020), with the background albedo extracted using the Joint Research Centre Two-stream Inversion Package (Pinty et al., 2011), thus without interannual variations and decoupled with simulated soil moisture (Fig. S1). The albedo of vegetated PFTs is prescribed with constant values in the model (Table S1). Since MICT is not capable of calculating leaves energy budgets separately from soils and stems, there's no vertical-layered temperature from the ground surface to the top of the canopy, and thus the $T_{surf}$ here is used to calculate all surface energy fluxes. The two heat fluxes out of the surface, i.e. H and LE are calculated following Eqs. (3) and (4), respectively:

$$H = \rho \times v \times C_d \times c_p \times (T_{surf} - T_{air}) \tag{3}$$

$$LE = L \times \beta \times \rho \times v \times C_d \times (q_{surf} - q_{air}) \tag{4}$$

where $\rho$ is air density (kg m$^{-3}$); v is horizontal wind speed (m s$^{-1}$); $C_d$ is drag coefficient (unitless); $c_p$ is the specific heat capacity of dry air (1004.675 J kg$^{-1}$ K$^{-1}$); $T_{air}$ is the air temperature (K); L is the latent heat of evaporation or sublimation (2.5008 or 2.8345 × 10$^6$ J kg$^{-1}$); $\beta$ is the limiting factor of potential total evapotranspiration (PET) (unitless); $q_{surf}$ and $q_{air}$ are the saturated moisture at the surface and in the air (kg kg$^{-1}$), respectively. Due to the differences in canopy height and leaf area index (LAI), $C_d$ should be different among different PFTs. To ensure compatibility with the grid-cell calculation of surface energy budget, the $C_d$ of different PFTs is weighted by their area. The LE (or ET) serves as the link between the energy cycle and hydrologic and carbon cycle. It is PFT-specific via a PFT-specific $\beta$ (see details in Ducharne, 2018) because the carbon cycle and hydrologic processes separate different PFTs or different soil tiles. In MICT, the LE consists of $E_{flood}$ (flood evaporation, not activated in the simulations of this study), $E_{subli}$ (snow sublimation), $E_{soil}$ (evaporation from bare soil), $E_{trans}$ (transpiration), and $E_{inter}$ (interception). Due to the distinct plant structures between different PFTs, the evaporation components

associated with vegetation, i.e. $E_{trans}$ and $E_{inter}$ are PFT-specific, while the $E_{subli}$ and $E_{soil}$ are calculated as the grid-cell average. For G, the energy exchange between surface and ground (snow or soil depending on the snow cover fraction), it is calculated following the classic Fourier's law (Hourdin, 1992). Considering that the calculation of G is identical to heat conduction in soil layers, the detailed derivation is only shown in Sect. 2.2 to avoid redundancy. For the areas covered by snow, the G also considers heat fluxes into snowpack which are used to melt snow.

## 2.2 Soil energy budgets

Table S2 displays the vertical discretization of soil in the current version of MICT. There are 32 soil layers for soil heat conduction with a total depth of 38 m and 11 soil layers for soil hydrology with a total depth of 2 m. These soil layers are located below the three snow layers in the model when there is snow (Fig. 1 (a)). As mentioned earlier, the heat conduction

across soil layers, snow layers, and between the surface and ground is calculated using the classic one-dimension Fourier's law (Hourdin, 1992), with the latent flux of soil-freezing taken into consideration (Gouttevin et al., 2012):

$$c\frac{\partial T_{soil}}{\partial t} = \frac{\partial}{\partial z}\left(\lambda \frac{\partial T_{soil}}{\partial z}\right) + \rho_{ice}L\frac{\partial \theta_{ice}}{\partial t} \tag{5}$$

where c is volumetric soil heat capacity (J $K^{-1}$ $m^{-3}$); $T_{soil}$ is the soil temperature (K); $\lambda$ is the soil thermal conductivity (J $m^{-1}$ $s^{-1}$ $K^{-1}$); $\rho_{ice}$ is ice density (920 kg $m^{-3}$); L is latent heat of fusion (0.3336 × $10^6$ J $kg^{-1}$); $\theta_{ice}$ is volumetric ice content ($m^3$ $m^{-3}$); t

is time (s) and z is soil layer depth (m). The c is calculated as the area-weighted sum of the heat capacity of liquid soil moisture (4.18 × $10^6$ J $K^{-1}$ $m^{-3}$), frozen soil moisture (2.11 × $10^6$ J $K^{-1}$ $m^{-3}$), and soil (depending on the soil type and the SOC content). The $\lambda$ is calculated following:

$$\lambda = Ke \times \lambda_{sat} + (1 - Ke) \times \lambda_{dry} \tag{6}$$

with

175 $$\lambda_{sat} = \lambda_{solid}^{(1-\theta_{sat})} + \lambda_{liq}^{\left(\theta_{sat} \times \frac{\theta_{liq}}{\theta_{liq} + \theta_{ice}}\right)} + \lambda_{ice}^{\left(\theta_{sat} \times \frac{\theta_{ice}}{\theta_{liq} + \theta_{ice}}\right)} \tag{7}$$

where $Ke$ is Kersten number calculated using a function of soil moisture saturated degree; $\lambda_{sat}$ and $\lambda_{dry}$ are the saturated and dry thermal conductivities (W $m^{-1}$ $k^{-1}$), respectively; $\theta_{sat}$ is saturated soil moisture, depending on the soil type and the SOC content ($m^3$ $m^{-3}$); $\theta_{liq}$ is volumetric liquid water content ($m^3$ $m^{-3}$); $\lambda_{solid}$ is the thermal conductivity of soil solid material, calculated as the geometric mean conductivities of mineral soil and SOC; $\lambda_{liq}$ and $\lambda_{ice}$ are the thermal conductivities of water

(0.57 W $m^{-1}$ $k^{-1}$) and ice (2.2 W $m^{-1}$ $k^{-1}$), respectively. The two thermal parameters are calculated for each soil layer because the heat is transferred vertically across all soil layers. The input liquid or frozen soil moisture (SM) are calculated as the area-weighted sum of all soil tiles to keep consistency with the grid-cell energy budget, despite the fact that soil hydrology and soil carbon processes operate for each sub-grid element (soil tile for hydrology or PFT for carbon).

## 2.3 Snow energy budgets

An explicit snow model of intermediate complexity has been introduced in MICT as described in Wang et al. (2013). The snowpack includes three snow layers (Fig. 1), with snow settling, water percolation, and refreezing and thawing of snow taken into account. When evaluated against observation data, the new snow module has improved the heat interaction between snow and soil or ground surface (Wang et al., 2013). The heat conduction in snow layers uses the same one-dimension heat diffusion function as that in soil layers (Eq. (5)), while with the snow's heat capacity ($c_{snow}$) and thermal conductivity ($\lambda_{snow}$):

$$c_{snow} = \rho_{snow} \times c_{p\_ice} \tag{8}$$

$$\lambda_{snow} = \left(a_\lambda + b_\lambda \times (\rho_{snow})^2\right) + \left(a_{\lambda v} + \frac{b_{\lambda v}}{T_{snow} - c_{\lambda v}}\right) \times \frac{P_0}{P_a} \tag{9}$$

where $\rho_{snow}$ is snow density (kg m$^{-3}$), varying with the snow settling; $c_{p\_ice}$ is the heat capacity of ice ($2.11 \times 10^6$ J K$^{-1}$ m$^{-3}$); the parameters $a_\lambda = 0.02$, $b_\lambda = 2.50 \times 10^{-6}$, $a_{\lambda v} = -0.06$, $b_{\lambda v} = -2.54$, $c_{\lambda v} = -289.99$, $P_0 = 1000$ hPa; $T_{snow}$ is the temperature of snow layers; and $P_a$ is atmospheric pressure (hPa). Besides the two thermal parameters, snow albedo ($\alpha_{snow}$) is a key variable affecting the surface energy budget (Wang et al., 2013). The value of $\alpha_{snow}$ is calculated following:

$$\alpha_{snow} = \alpha_{snow\_min} + k \times e^{\left(-\frac{age_{snow}}{\tau}\right)} \tag{10}$$

where $\alpha_{snow\_min}$ is the minimum snow albedo value after aging; k is the decay rate of snow albedo; $age_{snow}$ is the snow age; and $\tau$ is the time constant of the decay of snow albedo (10 days) (Chalita et al., 1994). Although $\alpha_{snow\_min}$ and k vary across different vegetation types (Table S3), the $\alpha_{snow}$ and all other snow-related processes including the heat conduction across the snow layers are still calculated at grid-cell scale by weighted area in MICT.

## 3 Implementation of tiling energy budgets in ORCHIDEE-MICT-teb

To represent the sub-grid energy budget in MICT, we calculate PFT-specific surface properties including the roughness height and the albedo of different PFTs to start the separation of surface energy budgets for each PFT, and then add the PFT-specific calculation for energy budget at the surface as well as in snow and soil layers (Figs. 1 and 2). Owing to using distinct input variables from the energy budget module, some processes in the hydrology cycle and carbon cycle are also modified correspondingly. The details of all of the modifications are described as follows.

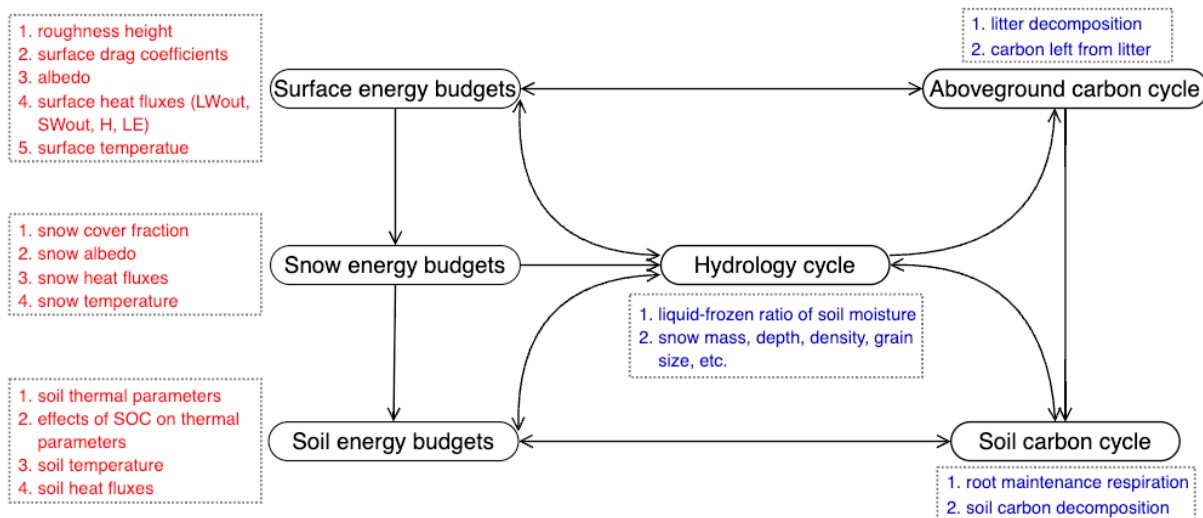

**Figure 2: Modifications of energy, hydrology, and carbon processes to implement tiling energy budgets in ORCHIDEE-MICT in this study. The black rounded rectangles show the main modules in ORCHIDEE-MICT, and the red or blue sharp rectangles show the processes which are modified to include the multi-tilling energy budget. The red and blue here distinct the processes where the PFT-specific calculations are added actively and those modified passively due to the input variables changing following the multi-tilling surface, snow, and soil thermics.**

### 3.1 Surface energy budgets

Differences in surface properties serve as the foundation for distinct surface energy budgets across sub-grids. For example, variations in vegetation heights among tree, grass, and bare soil result in different aerodynamic roughness, then drag
coefficients, and then different turbulent fluxes including H and LE (Eqs. (3) and (4)). Different surface types among tree, grass, and bare soil also result in different surface albedo ($\alpha$), influencing the amount of solar radiation reflected by the surface (Eq. (2)). In MICT-teb, we employ the specific roughness height ($H_{rough}$, vegetation height minus zero plane displacement height) and albedo for each PFT (Tables S2-S4 and Fig. S1), instead of the average values of all PFTs within a grid-cell used in MICT. That means, if only considering the changes in surface properties, the more heterogeneous the subgrid PFT
distribution is, the larger differences in the energy and the subsequent hydrology and carbon-related processes between MICT-teb and MICT may be. The distinct surface properties among different PFTs propagate to differences in surface energy fluxes and then differentiate the $T_{surf}$ across different PFTs. In MICT-teb, all processes related to the changes in surface properties and $T_{surf}$ in the surface energy budget are separated and operate independently for each PFT (Fig. 2).

For LE (or ET), it comprises four components: $E_{subli}$, $E_{soil}$, $E_{trans}$, and $E_{inter}$ in the model. As described in Sect. 2.1, only $E_{trans}$ and $E_{inter}$ are PFT-specific in the original MICT. However, due to the modifications made to surface energy budgets and surface properties resulting in varying snow cover fractions across different PFTs, $E_{subli}$ is now separated for each PFT in MICT-teb.

Regarding $E_{soil}$, the water limitation in the hydrology module differs among different soil tiles, whereas in MICT, it is considered at grid scale only. The calculation of $E_{soil}$ has been modified by using the specific water limitation for each soil tile in MICT-teb. All PFTs in one soil tile are distributed the same value of $E_{soil}$, and then are used in PFT-specific calculation in energy modules.

## 3.2 Snow energy budgets

The modifications to the surface energy budget in MICT-teb, which cause variations in heat fluxes into the snowpack, result in differences in the formation and melting of snow and then snow mass, snow depth ($dz_{snow}$), and snow density ($\rho_{snow}$) among different PFTs. The snow cover fraction ($f_{snow}$) as calculated in MICT follows (Niu and Yang, 2007; Wang et al., 2015):

$$f_{snow} = tanh(\textstyle\sum_{i=1}^{3}(dz_{snow,i})/0.025 * (\textstyle\sum_{i=1}^{3}(dz_{snow,i} * \rho_{snow,i}) / \textstyle\sum_{i=1}^{3}(dz_{snow,i}))/50) \tag{11}$$

would be different among different PFTs, where i is the index of snow layer. The variations in $f_{snow}$ would subsequently influence the surface albedo due to snow (Eq. (10)) and, in turn, the snow feedbacks on the surface energy budget. All of the snow-related processes have been separated for each PFT in MICT-teb (Fig. 2).

## 3.3 Soil energy budgets

Following Eq. (5), modifications of energy budgets at surface and in snow layers could result in variations in the starting point of heat conduction in soil for different PFTs. While within soil, the heat conduction is more regulated by the heat capacity (c) and thermal conductivity ($\lambda$). Liquid SM, frozen SM, and SOC are three key factors influencing the two thermal parameters in the model (Eqs. (6) and (7)). In the original MICT, the average values of these three factors across all PFTs are used due to the limitation of having a grid-scale mean energy budget. In the new version, we simulate PFT-specific liquid SM and frozen SM in energy modules to represent the heterogeneity of different PFTs, following the separation of soil heat conduction (Fig. 2). Regarding SOC, MICT uses the grid-cell SOC obtained from an observation-based SOC map (FAO, 2012; Zhu et al., 2019; Guimberteau et al., 2018; Hugelius et al., 2013) and therefore the effects of SOC on thermal parameters are represented homogeneously within each grid cell. According to Zhu et al. (2019), an increase of 20 kg C $m^{-3}$ in SOC between two PFTs, which can be found in site-level data (Palmtag et al., 2022) and is reproduced in model simulations (Fig. S2), could result in a 42-52% decrease in thermal diffusivity ($\lambda$/c) and a subsequent 13-18% increase in current permafrost extent. The important role of SOC in regulating soil thermal regimes and the heterogeneity of SOC across different PFTs highlight the pressing need to represent the thermal effects of SOC for each sub-grid. However, limited by the availability of observed SOC data that could be prescribed for sub-grids at the regional or global scale, we utilize the simulated SOC for each PFT in MICT-teb and the simulated total SOC of all PFTs in MICT. Using simulated instead of prescribed SOC has the advantage of making the modeled SOC fully consistent with the simulated soil physics, but it has the drawback that SOC formed by processes that cannot be

simulated by the model (e.g. Pleistocene ecosystems such as Yedoma, thermokarst lakes filling by organic sediments) will be ignored, causing a possible mismatch with observed SOC density. Nevertheless, a comparable spatial pattern of gridded SOC for 0-3 m over the Northern Hemisphere (NH) against observation-based SOC data, as well as a comparable vertical profile against site-level data (Palmtag et al., 2022), confirm the model's ability to simulate the total volume (Fig. S3) and the PFT-specific vertical profiles (Fig. S2) of SOC.

## 3.4 Associated modifications of hydrological and carbon processes

Soil temperature ($T_{soil}$) is a key factor to influence hydrologic and biogeochemical processes in soil. Therefore, the PFT-specific variations in $T_{soil}$ result in a series of associated modifications of hydrological and carbon-related processes in relation to the original MICT (Fig. 2). For soil hydrology, there are three main modifications in MICT-teb compared to MICT: 1) the calculation of PFT-specific $T_{soil}$ for each PFT to calculate liquid-frozen ratio of SM; 2) the use of PFT-specific bare soil evaporation from the energy module; and 3) the separation of snow-related processes for each PFT. For the soil carbon cycle, there are four main modifications in MICT-teb: 1) the use of PFT-specific $T_{soil}$ and SM for litter decomposition; 2) the use of PFT-specific $T_{soil}$ and SM to calculate carbon flow from litter to soil; 3) the use of PFT-specific $T_{soil}$ and SM for root maintenance respiration; and 4) the use of PFT-specific $T_{soil}$ and SM for soil carbon decomposition.

## 4 Simulation protocol and forcing datasets

To compare the differences in energy, hydrology, and carbon processes between MICT-teb and MICT, we design three groups of simulations (S0, S1, and S2) as shown in Table 1. All of the three groups are run for the NH (0°-90°N) at a spatial resolution of 2° × 2°, with four simulation periods: A) Spin-up1, 100 years of the full ORCHIDEE with a looped 1901-1920 climate, $CO_2$ level of 1901 at 296.80 ppm, and the land cover map of 1901; B) SubC, 10,000 years of the soil carbon sub-model to accumulate SOC; C) Spin-up2, 50 years of the full ORCHIDEE to reach equilibrium with a looped 1901-1920 climate, $CO_2$ level of 1901, and the land cover map of 1901; and D) Transient simulation, the full ORCHIDEE with varying climate, $CO_2$ level and land cover maps from 1901 to 2020. The climate forcing data are obtained from CRU-JRA v2.3 (the version used in Global Carbon Budget 2022) (Friedlingstein et al., 2022), while the land cover maps are generated by combining the land cover map from TRENDY for 15 PFTs (bare soil, 8 tree PFTs, 4 grass PFTs, and 2 crop PFTs) (Lurton et al., 2020) and the peat map from Xu et al. (2018) for the peat grass PFT (Xu et al., 2018; Qiu et al., 2019). In S0, the original MICT is used for all four periods. In S1, MICT-teb is used with the flags controlling the tiling energy budget (TEB) turned off for periods A and B (i.e., identical to group S0), but turned on for period C. In this way, the differences in energy, hydrology, and carbon between S0 and S1 solely due to the TEB can be compared based on the same starting point (end of period B). In S2, the flags controlling the TEB

are turned on from the beginning of period A. Thus, comparing S0 and S2 could infer differences in long-term equilibrium of

energy and hydrology, as well as near-equilibrium soil carbon storage between MICT-teb and MICT.

**Table 1. Simulation protocol. MICT and MICT-teb indicate the original ORCHIDEE-MICT (without tiling energy budget) and the new ORCHIDEE-MICT-teb (with tiling energy budget), respectively. OFF and ON indicate turning off and turning on the flags which control the tiling energy budget in MICT-teb, respectively. If the flags are turned off, all of the PFT-specific variables related**

**to energy budget will use the grid-cell mean value in MICT-teb.**

| Simulation | | S0 | S1 | S2 |
|---|---|---|---|---|
| Model version | | MICT | MICT-teb | MICT-teb |
| Period | A | √ | √ (OFF) | √ (ON) |
| | B | √ | √ (OFF) | √ (ON) |
| | C | √ | √ (ON) | √ (ON) |
| | D | √ | | √ (ON) |
| Notes for the period | A | Spin-up1 (100 yr), Climate: cycle 1901-1920, $CO_2$: 1901, LUC: 1901 | | |
| | B | SubC (10,000 yr) | | |
| | C | Spin-up2 (50 yr), Climate: cycle 1901-1920, $CO_2$: 1901, LUC: 1901 | | |
| | D | Transient simulation, Climate: 1901-2020, $CO_2$: 1901-2020, LUC: 1901-2020 | | |

## 5 Evaluation of the impacts of tiling energy budget on energy, hydrology and carbon processes

Following the description of the simulation protocol in Sect. 4, this section presents the differences in energy, hydrology, and carbon processes between MICT-teb and MICT. Sect. 5.1 presents the comparison of S1 and S0, i.e. the impacts solely due to

TEB, while Sect. 5.2 presents the comparison of all the three simulations across the first three simulation periods, i.e. the long-term impacts of TEB on energy, hydrology, and carbon processes. Unless otherwise stated, all differences indicate the mean values of the last ten years in period C from MICT-teb minus those from MICT in Sect. 5.1.

### 5.1 Impacts solely due to tiling energy budget

### 5.1.1 Surface energy budgets

To explain the differences in energy budgets between MICT-teb and MICT, we begin our analysis by randomly selecting three grid-cells at latitudes for tropical (17ºN, 155ºW), temperate (51ºN, 101ºW) and boreal (71ºN, 147ºE) biomes (Fig. 3). For the

energy budgets at surface (Eq. (2)), the $SW_{in}$ and $LW_{in}$ are the same between the two versions (not shown) because both variables come from the input climate data, while the other four main surface energy fluxes including $SW_{out}$, $LW_{out}$, H, and

LE show significant differences due to TEB. The differences in energy fluxes can be well explained by the differences in surface properties at all three grid-cells for the different latitudes. For instance, the difference in $SW_{out}$ between MICT-teb and MICT can be explained by the difference in albedo; the difference in $LW_{out}$ can be explained by the difference in $T_{surf}$; the difference in H (or LE) can be explained by the difference in $T_{surf}$ and / or $C_d$ (surface drag coefficients). This correlation agrees well with theoretical equations (Eqs. (2)-(4)).


The variations in surface properties are related to the modifications made to represent the PFT-specific information in MICT-teb. Regarding albedo, grass leaves generally have a higher albedo (0.15-0.16) than tree leaves (0.10-0.14) (Table S1), while the albedo of bare soil varies, in the model, depending on soil moisture, ranging from ~0.05 in moist areas to ~0.5 in dry areas (Fig. S1). In the tropical grid-cell, all of the four grass PFTs show a higher albedo than the grid-cell mean (0.125), whereas

some of the six tree PFTs show a lower albedo than the grid-cell mean and others showing a higher value. The peat PFT has the same leaf albedo value as grass PFTs, but its albedo, as shown in Fig. 3b, is lower than the grid-cell mean, which is due to the higher fraction of bare soil (with a small albedo of 0.093) for this PFT (~70%) relative to the other four grass PFTs (2-30%). We note that the cover fraction of a PFT in the model includes both the leaf-covered area (the canopy) and the no-leaf-covered area (the soil), depending on a function of leaf area index. The albedo of a PFT is calculated as the area-weighted sum

of the albedo of leaves and the albedo of soil within this PFT. In temperate and boreal regions, the albedo of one PFT is greatly influenced by snow cover fraction owing to the significantly higher albedo of snow (Table S3). Consequently, the pattern of the difference in albedo between MICT-teb and MICT (Figs. 3 (g) and (l)), closely resembles the difference in snow cover fraction (Fig. S4) for temperate and boreal grid-cells. Regarding $C_d$, the surface drag coefficient, its variations are determined by variations in PFT-specific $T_{surf}$ and $H_{rough}$ (roughness height): the smaller the $T_{surf}$ and the larger the $H_{rough}$, the larger the $C_d$.

The theoretical relationship can be reproduced well from the comparison of simulated results between MICT-teb and MICT in Fig. 3.

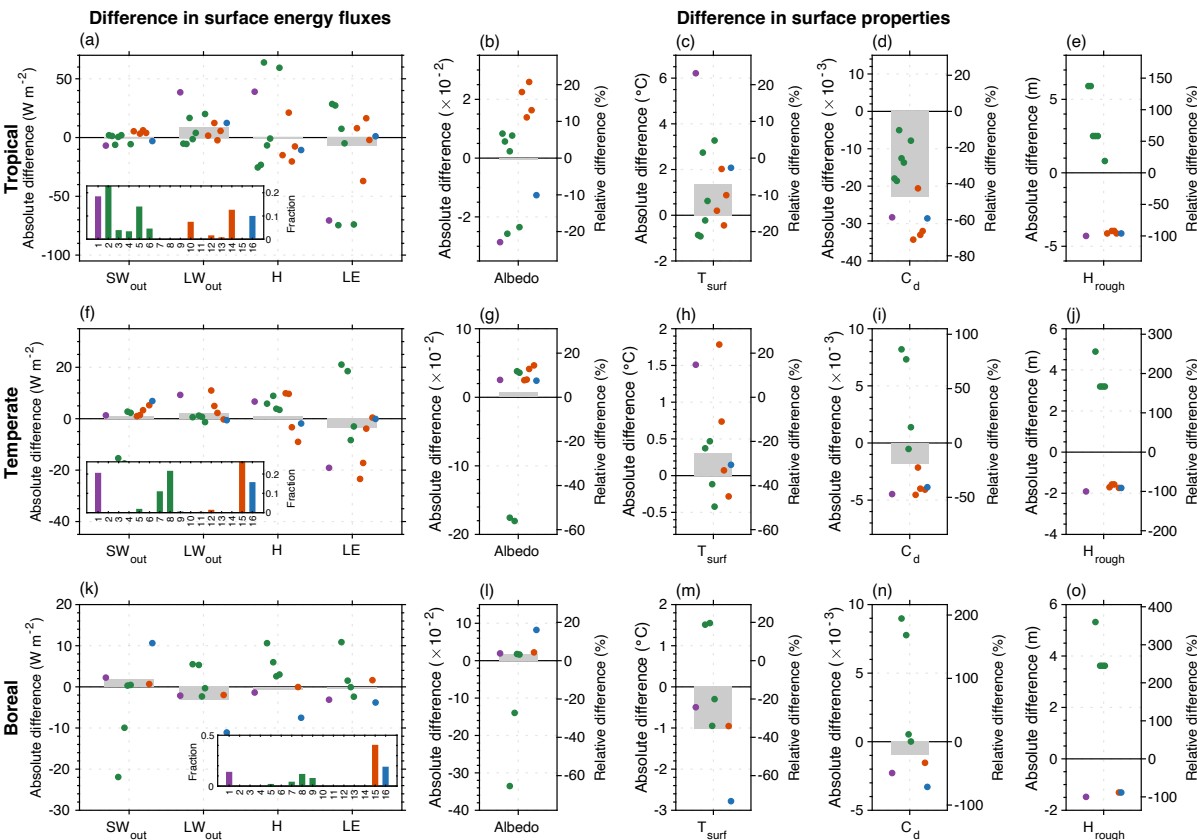

**Figure 3: Differences in surface energy fluxes (the first column) and surface properties (from the second to the last column) between MICT-teb and MICT for three grid-cells located at tropical (17ºN, 155ºW), temperate (51ºN, 101ºW), and boreal (71ºN, 147ºE) regions, respectively. The surface energy fluxes include outward shortwave radiation (SW_out), outward longwave radiation (LW_out), sensible heat flux (H), and latent heat flux (LE). The surface properties include albedo (Albedo), surface temperature (T_surf), surface drag coefficient (C_d), and roughness height (H_rough). The gray bar in the background indicates the grid-scale difference in each variable. The colored points from left to right indicate the differences between 16 PFTs in MICT-teb and the grid-scale values in MICT, and the missing points indicate the cover fraction of one PFT is zero for the grid-cell. The insets in the first column show the cover fraction of 16 PFTs for the grid-cell. The 16 PFTs are bare soil (PFT1, in purple), trees (PFT 2-9, in green), grass (PFT 10-15, in orange), and peat grass (PFT16, in blue). Please see Table S3 for the long name of 16 PFTs.**

When extending to the whole NH, the correspondence between differences in surface heat fluxes and differences in surface properties at grid-cell scale can still be observed (Figs. 4-5 and S5). Additionally, certain latitudinal trends begin to emerge in the difference between MICT-teb and MICT. Based on our modifications to calculate PFT-specific leaf albedo and H_rough, the most immediate variations in SW_out or turbulent fluxes (H and LE) lead to the final direction of differences in T_surf between MICT-teb and MICT for different PFTs (Table 2). For bare soil, whose H_rough is set to 0 in MICT-teb, the smaller H and LE result in a higher T_surf (up to 3 °C) compared to MICT across almost all areas with bare soil. For tree PFTs, the higher H and LE due to the larger H_rough contributes overall to a cooler T_surf (-3-0 °C) at low latitudes, while the more important decrease in SW_out due to the smaller albedo in MCT-teb results in a warmer T_surf (0-2 °C) at high latitudes. With the same dominant role

of the $H_{rough}$ variation at low latitudes and of the albedo variation at high latitudes, the grass and peat grass show a warmer $T_{surf}$ (0-3 °C) at low latitudes but a cooler $T_{surf}$ (-1-0 °C for grass and -2-0 °C for peat grass) at high latitudes in MICT-teb. As a result, the grid-cell $T_{surf}$ simulated by MICT-teb is 0-2°C higher in most regions in the NH while slightly cooler (-1-0°C) in

the north of 60°N and some arid regions than MICT (Fig. 5b).

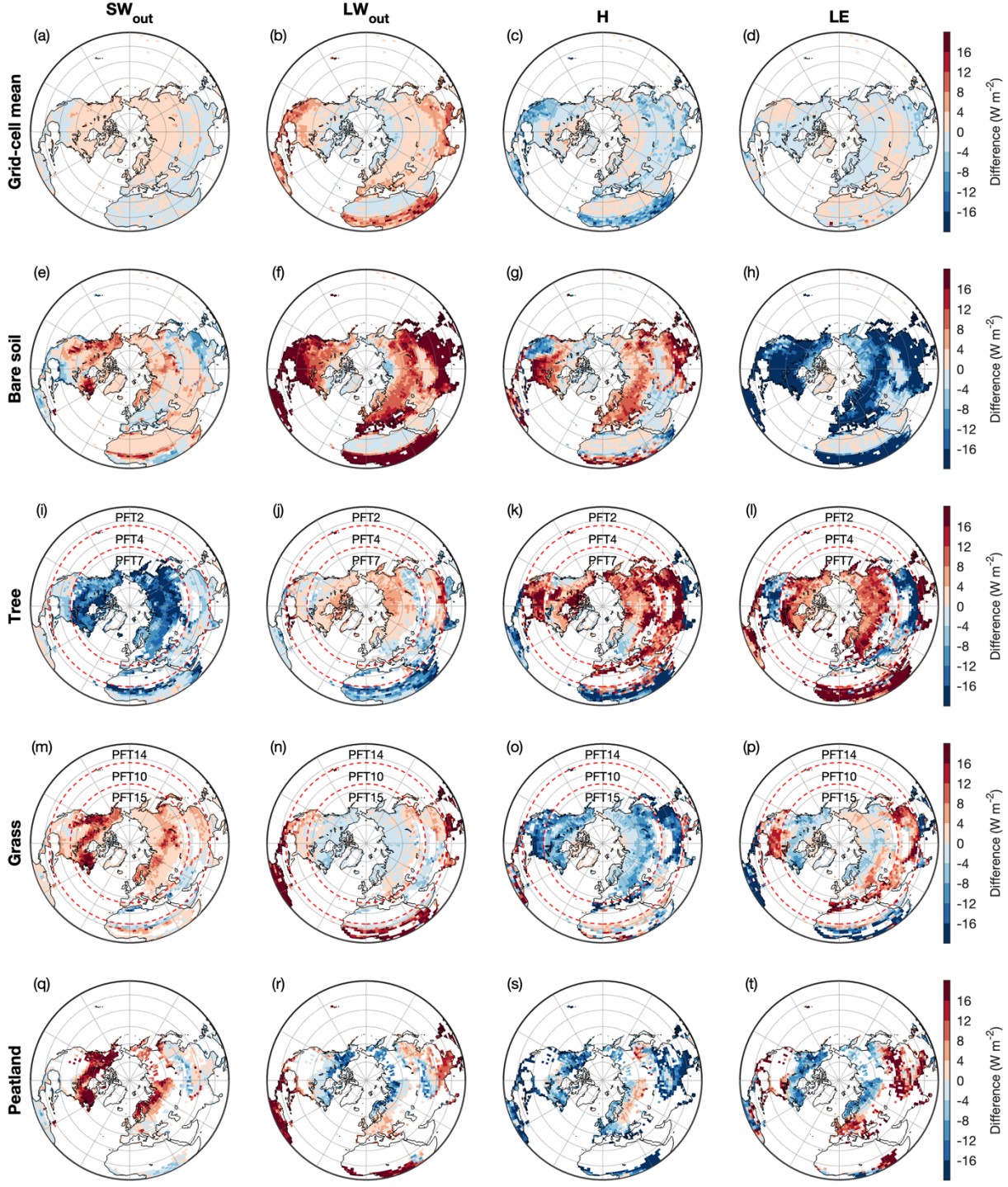

**Figure 4: Spatial patterns of differences in surface energy fluxes including outward shortwave radiation (SW_out), outward longwave radiation (LW_out), sensible heat flux (H), and latent heat flux (LE) between MICT-teb and MICT over the Northern Hemisphere. The first to fifth lines show the difference in each flux between the grid-cell mean or four tiles from MICT-teb and the grid-cell mean**

**from MICT, respectively. The four tiles include bare soil (PFT1), tree (a combination of PFT2 (Tropical broad-leaved evergreen tree) in the south of 20ºN, PFT4 (Temperate needleleaf evergreen) between 20ºN and 40ºN and PFT7 (Boreal needleleaf evergreen tree) in the north of 40ºN), grass (a combination of PFT14 (Topical C3 grass) in the south of 20ºN, PFT10 (Temperate C3 grass) between 20ºN and 40ºN, and PFT15 (Boreal C3 grass) in the north of 40ºN), and peatland grass (PFT16). The three PFTs for the tree or grass are combined just in order to show as many results as possible, and only one PFT is shown in each grid-cell.**


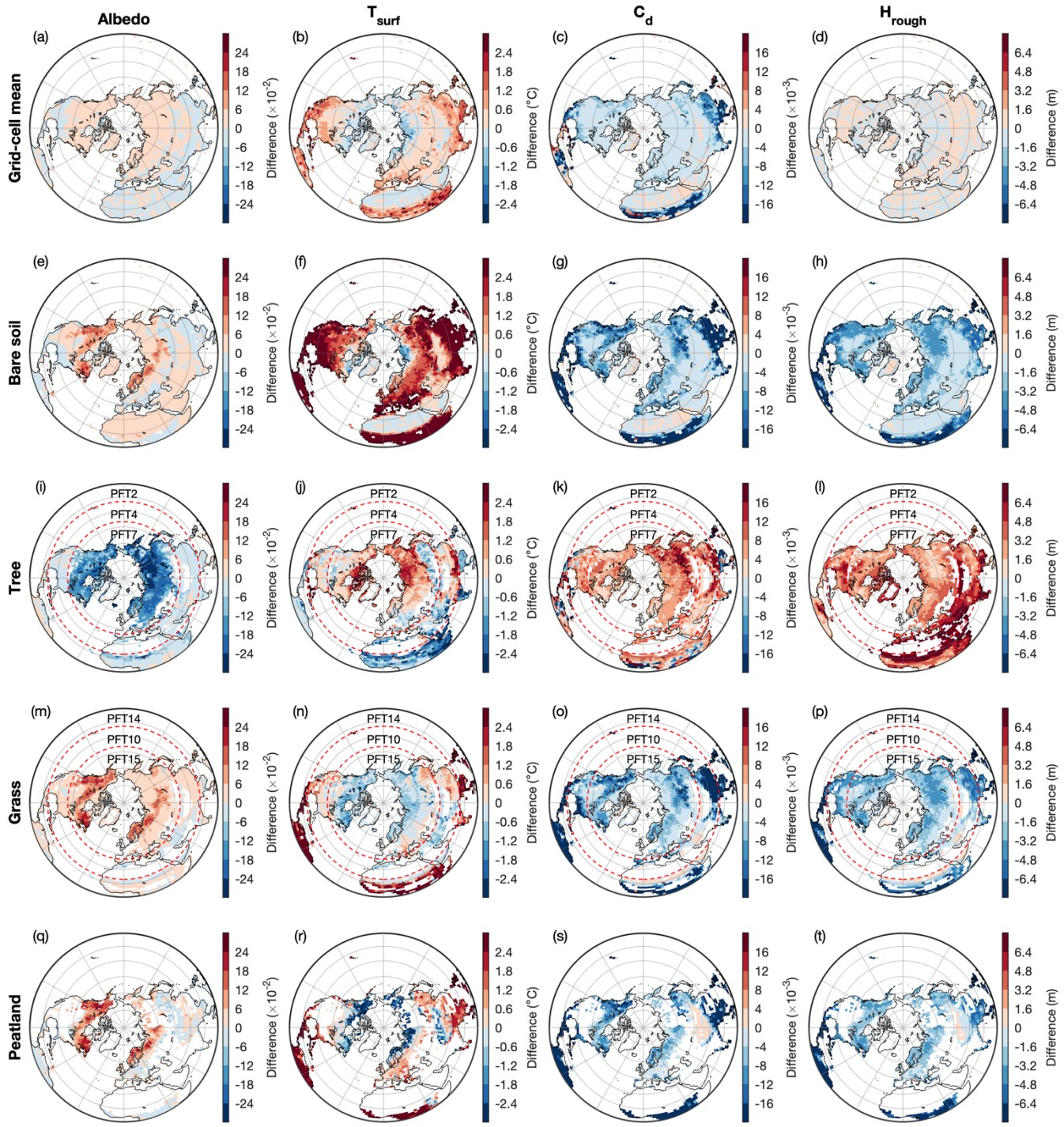

**Figure 5: Same as Fig. 4, but for spatial patterns of differences in surface properties including albedo (Albedo), surface temperature ($T_{surf}$), surface drag coefficients ($C_d$), and roughness height ($H_{rough}$) between MICT-teb and MICT over the Northern Hemisphere.**

**Table 2. Qualitative summary of latitudinal trends of differences in surface properties and associated differences in surface energy fluxes between MICT-teb and MICT. The red and blue arrows indicate the warming and cooling effects on surface temperature, respectively. The two arrows indicate a stronger effect than one arrow.**

| PFT type | Regions | ΔAlbedo | ΔSW$_{out}$ | ΔH$_{rough}$ | ΔH and ΔLE | ΔT$_{surf}$ |
|---|---|---|---|---|---|---|
| **Bare soil** | Low / High-latitudes | … | … | ↓ | ↓↓ (red) | 🌡 (red) |
| **Tree** | Low-latitudes | ↓ | ↓ (red) | ↑ | ↑↑ (blue) | 🌡 (blue) |
| | High-latitudes | ↓ | ↓↓ (red) | ↑ | ↑ (blue) | 🌡 (red) |
| **Grass** | Low-latitudes | ↑ | ↑ (blue) | ↓ | ↓↓ (red) | 🌡 (red) |
| | High-latitudes | ↑ | ↑↑ (blue) | ↓ | ↓ (red) | 🌡 (blue) |
| **Peat grass** | Low-latitudes | ↑ | ↑ (blue) | ↓ | ↓↓ (red) | 🌡 (red) |
| | High-latitudes | ↑ | ↑↑ (blue) | ↓ | ↓ (red) | 🌡 (blue) |

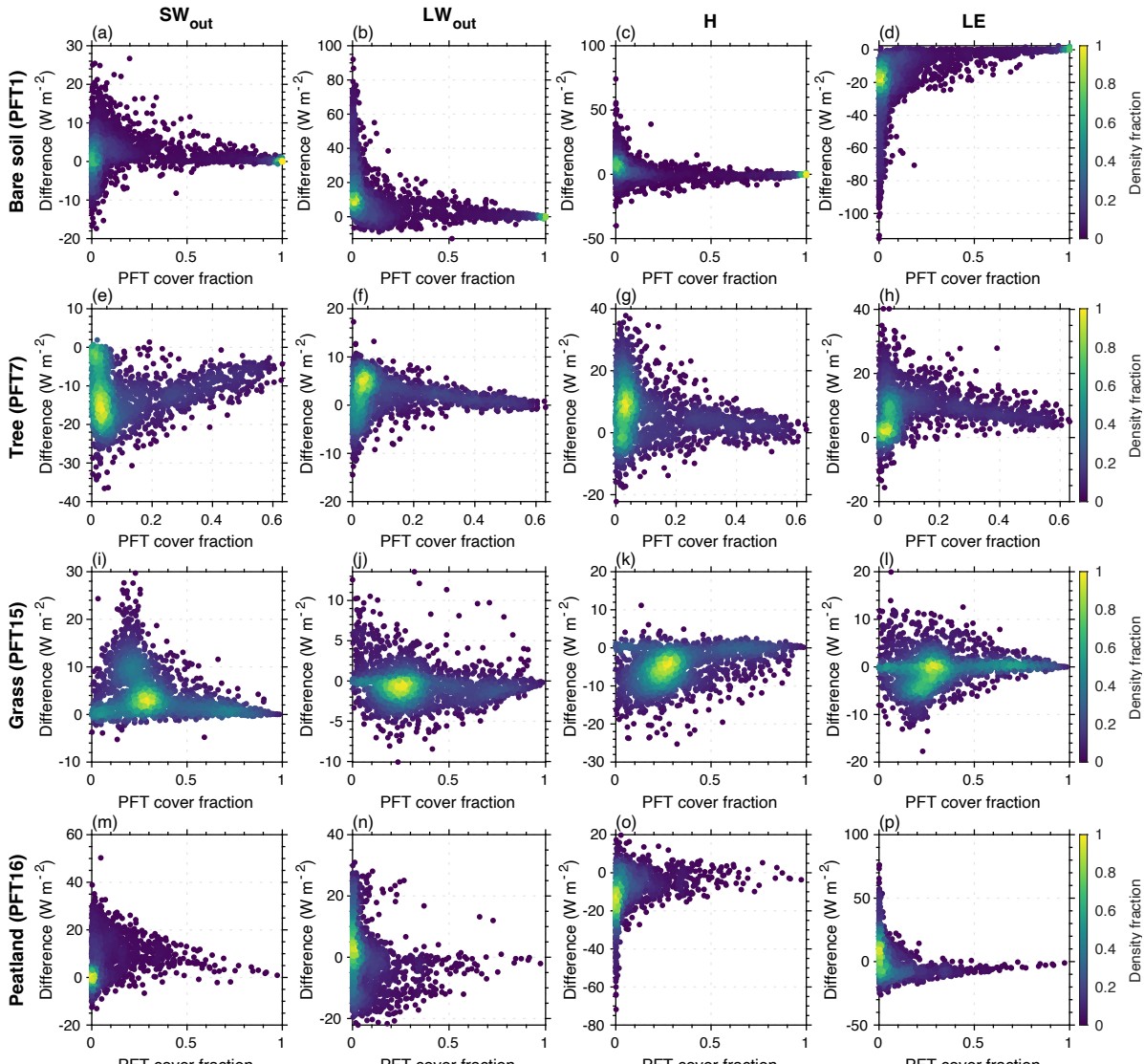

**Figure 6: Scatter plot of differences in surface energy fluxes between MICT-teb and MICT versus vegetation cover fraction for bare soil (PFT1), tree (PFT7), grass (PFT15), and peatland grass (PFT16). The surface energy fluxes include outward shortwave radiation (SW_out), outward longwave radiation (LW_out), sensible heat flux (H), and latent heat flux (LE). The color of each point represents the density fraction of grid-cells.**

Another interesting result is the relationship between differences in surface heat fluxes or properties between MICT-teb and MICT and the vegetation cover fraction. As mentioned in Sect. 3.1, the difference between the two versions should become smaller where one PFT's tends to become more dominant in the grid-cell. When the cover fraction of one PFT approaches 100%, there will be no difference between the grid-cell mean and the specific PFT. Taking four PFTs as examples, we found this pattern both for surface heat fluxes (Fig. 6) and surface property variables (Fig. S6).

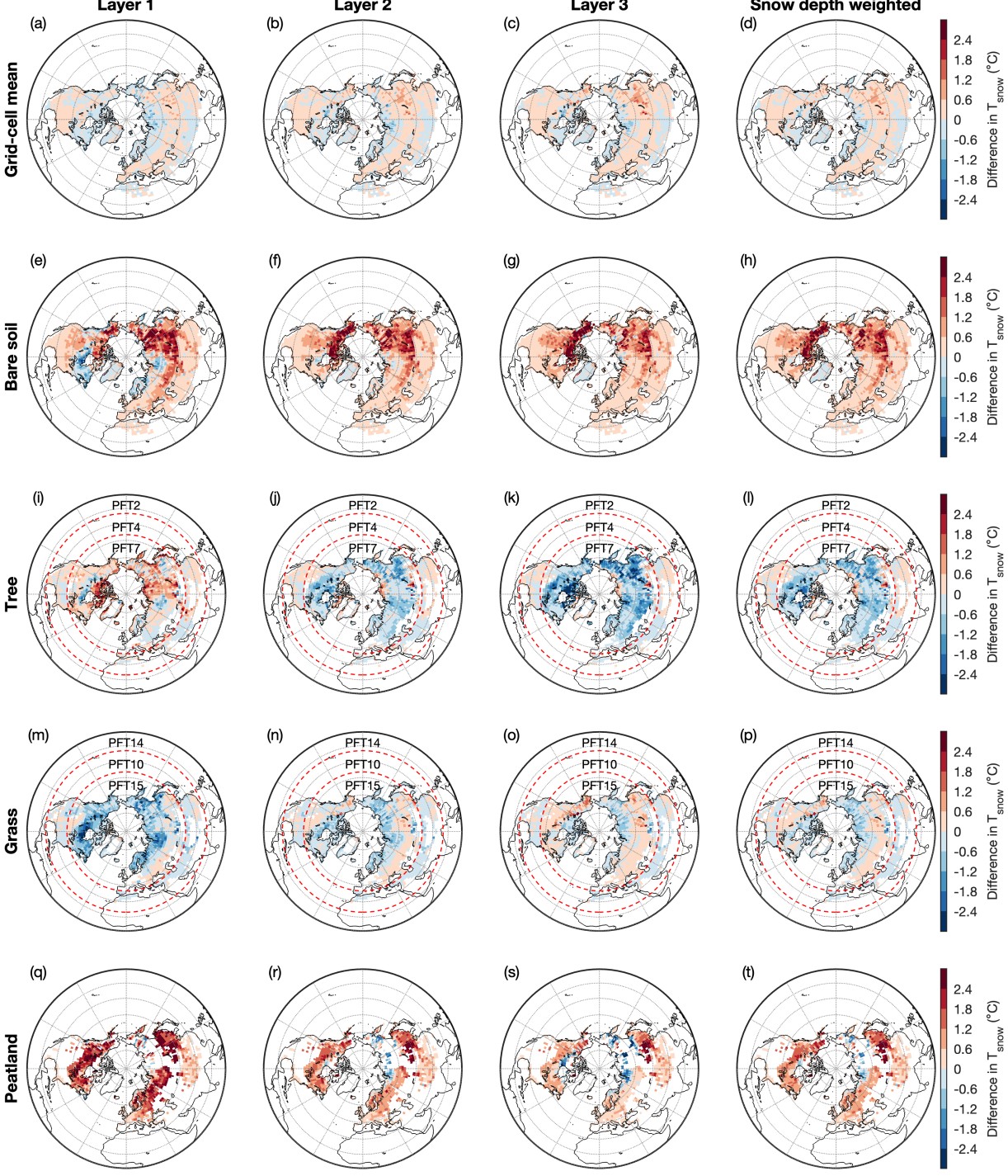

**Figure 7: Spatial patterns of difference in $T_{snow}$ between MICT-teb and MICT for three snow layers and the snow depth weighted results over the Northern Hemisphere. The snow layers from up to down are numbered 1-3, respectively. The snow depth weighted results are shown due to the different snow layer depth across different grid-cells.**


### 5.1.2 Snow energy budgets

Figure 7 presents differences in $T_{snow}$ ($\Delta T_{snow}$) of three snow layers between MICT-teb and MICT for the grid-cell mean and four PFT types. Overall, the grid-cell $\Delta T_{snow}$ follows the $\Delta T_{surf}$ between the two versions, with an up to 1 °C warmer snow layer across most areas in MICT-teb. The correlation of $\Delta T_{snow}$ and $\Delta T_{surf}$ weakens from the uppermost snow layer ($T_{snow,1}$) to

the bottom one ($T_{snow,3}$), especially for tree and grass PFTs (Table S5). For the two PFTs, the differences in the two thermal parameters ($c_{snow}$ and $\lambda_{snow}$) play a more important role than $\Delta T_{surf}$ in shaping the spatial pattern of $\Delta T_{snow}$ in the bottom two layers (Table S5 and Figs. S7 and S8). Since the snow depth for each layer is not fixed like soil layers (Fig. 1), we calculate the snow-depth weighted $\Delta T_{snow}$ between the two versions (the last column in Fig. 7). The differences in $c_{snow}$ and $\lambda_{snow}$ are more important in determining the spatial patterns of the snow-depth weighted $\Delta T_{snow}$.


### 5.1.3 Soil energy budgets

Figure 8 presents the differences in $T_{soil}$ ($\Delta T_{soil}$) of four soil layers between MICT-teb and MICT for the grid-cell mean and four PFT types. Similar to the $\Delta T_{snow}$, the $\Delta T_{soil}$ shows a larger and significantly positive correlation (R = 0.31-1.00, p < 0.05) with differences in the starting point of heat conduction ($T_{surf}$ for 0°-30°N and $T_{snow,3}$ for 30°N-90°N) than the two thermal

parameters of soil (Table S6). The grid-cell mean $T_{soil}$ for the four soil layers simulated by MICT-teb is ~0.6 °C warmer than MICT in the north of 30°N while ~1.2 °C warmer in the tropics across four soil layers. The PFT-specific $\Delta T_{soil}$ show considerably different magnitudes and directions across four PFT types: the $T_{soil}$ for bare soil is ~3 °C higher in MICT-teb than MICT; the $T_{soil}$ for tree and peat grass is 0.6-3 °C lower; and the $T_{soil}$ for grass is 0.6-3 °C higher. Despite using the same parameter values of leaf albedo and $H_{rough}$ between peat grass and C3 grass, the $T_{soil}$ of peat grass is 0.6-3 °C lower at high

latitudes in MICT-teb, which could be related to the considerably different spatial patterns of the two soil thermal parameters between peat grass and C3 grass (Table S6 and Figs. S9 and S10).

As mentioned in Sect. 3.3, the liquid SM ($SM_{liquid}$), frozen SM ($SM_{frozen}$), and SOC are three key factors influencing the two thermal parameters of soil (Eqs. (6) and (7)). Despite the previous soil-tile-based hydrologic processes and PFT-based carbon

cycle in MICT, the soil thermal parameters (c and $\lambda$) are calculated with the grid-cell mean values of $SM_{liquid}$, $SM_{frozen}$, and SOC in that version. With far wetter SM (≥200%, Figs. S11 and S12) and far more SOC storage (≥200%, Fig. S13), the peat PFT has a ~200% higher c (Fig. S9) and a ~100% higher $\lambda$ (Fig. S10) than grid-cell mean. Such large $\Delta c$ and $\Delta\lambda$ compared to grass (40% higher / lower c and 20% higher / lower $\lambda$ than grid-cell mean) leads to a lower $T_{soil}$ for peat grass in MICT-teb

than in MICT (Fig. 8). Besides the peat grass PFT, the important role of the three factors in regulating $T_{soil}$ can also be found

for other PFTs. For example, we found ~100% less SOC storage in bare soil than the grid-cell mean, which contributes to the higher $T_{soil}$ for bare soil due to the absence of SOC's insulating impacts.

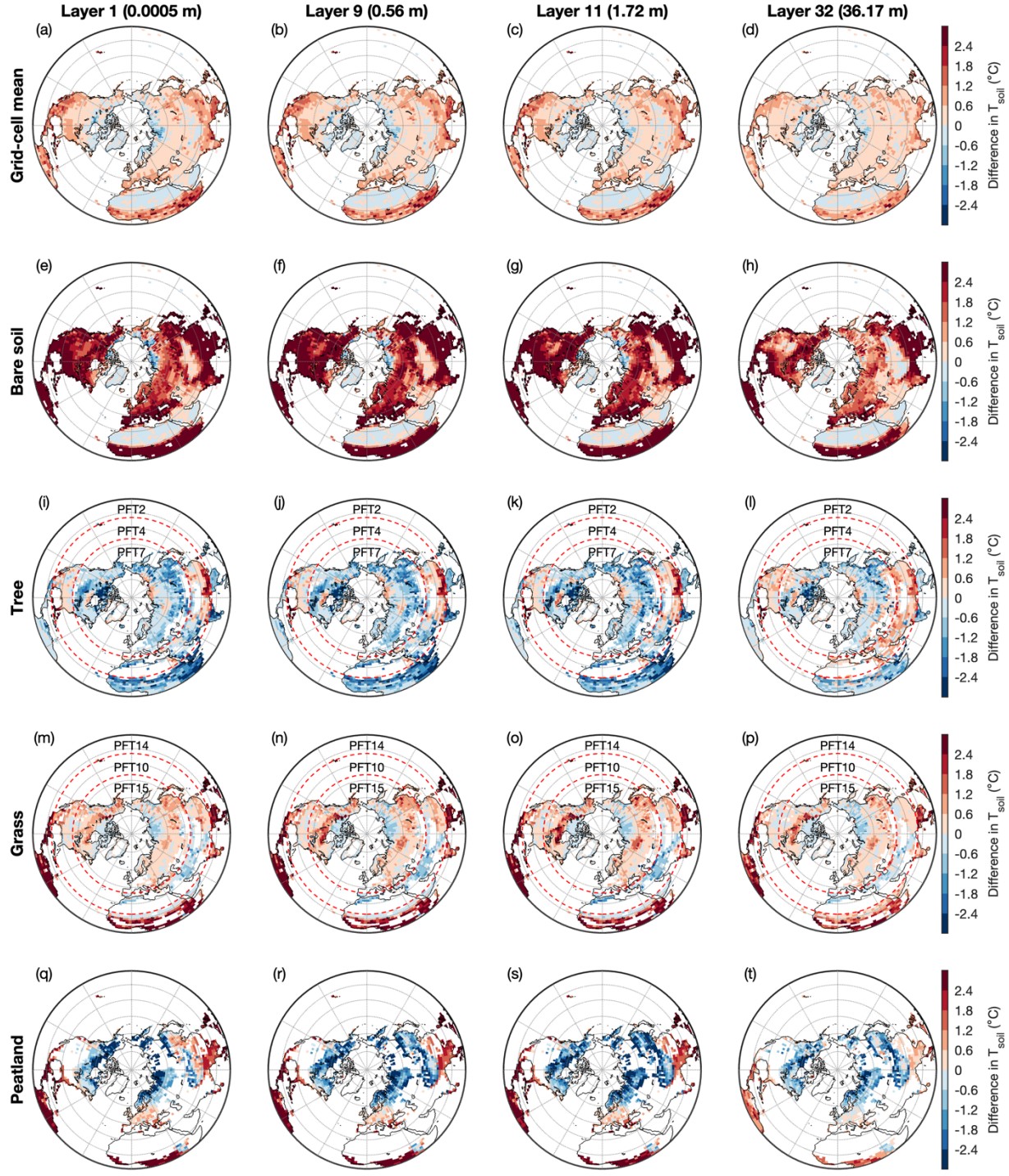

**Figure 8: Same as Fig. 7, but for spatial patterns of differences in T$_{soil}$ between MICT-teb and MICT for four soil layers over the Northern Hemisphere.**

## 5.2 Long-term impacts on energy, hydrology, and carbon cycle

### 5.2.1 Energy

Regarding the long-term impacts of representing the sub-grid energy budgets in the model, we compare the simulated results over the NH among S0, S1, and S2 from three aspects: energy budgets, hydrology, and carbon cycle (Figs. 9-11). For the energy budgets, we found that the difference in $T_{surf}$ over the NH between S1 and S0, or between S2 and S0 appears in the first two to three years and then remains stable throughout all three simulation periods (Fig. 9a). By the end of period C, the $\Delta T_{surf}$ over the NH between S2 and S0 remains at 0.37 °C (+3.5%). A very similar $\Delta T_{surf}$ (0.38 °C, 3.6%) can be observed between S1 and S0, suggesting that the surface energy budgets can quickly respond to variations in surface properties including albedo and roughness. As the heat moves down, the determining role of $T_{surf}$ in influencing the heat conduction in soil (Table S6) makes the $T_{soil}$ at the 1st (0.0005 m), 11st (1.72 m), and 32nd (36.17 m) layers over the NH warmer by 0.44 °C (3.3%), 0.45 °C (3.5%) and 0.51 °C (3.6%) respectively, by the end of period C under S2 than that under S0. But it takes a longer time to reach stability for $T_{soil}$ at the bottom soil layers than the upper ones (Fig. 9 (c), (e), and (g)). The SOC, acting as an insulator, could regulate the soil thermics, especially in summer (Zhu et al., 2019). Nevertheless, the difference in mean annual or monthly $T_{soil}$ at all soil layers between S1 and S2 is very small, no more than 0.18 °C (1.2%). Therefore, the long-term effects of TEB on surface or soil energy budgets over the NH is subtle in the simulations of this study.

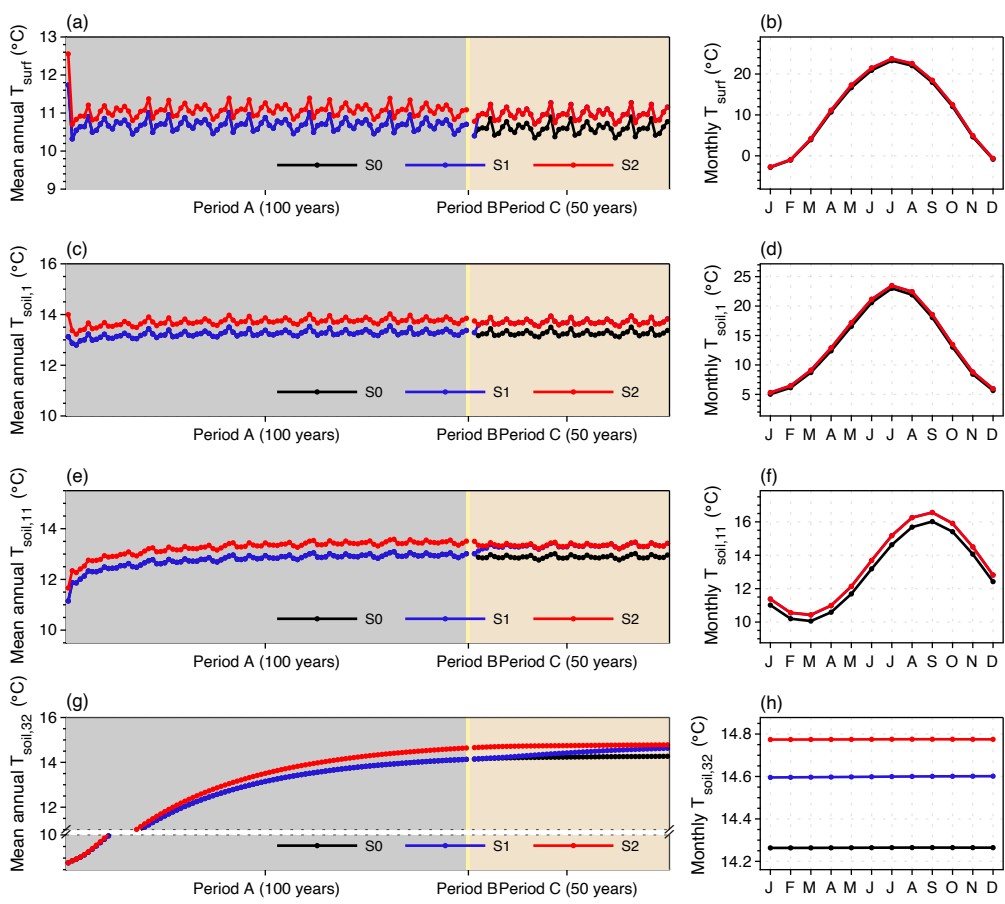

**Figure 9: Time series and seasonal cycle of surface temperature ($T_{surf}$) and soil temperature for three soil layers ($T_{soil,1}$, $T_{soil,11}$, and $T_{soil,32}$) over the Northern Hemisphere from three simulations. The annual values are calculated using the yearly average, and the monthly values are calculated using the average of the last ten years in period C. The backgrounds in Figs. (a), (c), (e), and (g) are colored to help explain the simulation length of the three periods. Detailed explanations for the three simulations (S0, S1, and S2) and the three periods (A, B, and C) can be found in Table 1 and Sect. 4.**

### 5.2.2 Hydrology

The hydrology processes over the NH can respond to the representation of sub-grid energy budgets as quickly as temperature, taking up to five years to reach the stability for both surface water fluxes including evapotranspiration (ET) and surface runoff (Q), as well as subsurface water fluxes such as drainage (D) (Fig. 10). Overall, MICT-teb shows a smaller ET (-8.7 mm yr$^{-1}$, -2.3%), but a larger D (+11.8 mm yr$^{-1}$, 8.3%) over the NH compared to MICT. When separating sub-components of ET and soil tiles, we found that the decreased ET is mainly contributed by the decreased $E_{trans}$ (transpiration) from tree PFTs and the decreased $E_{subli}$ (sublimation) from the grass PFTs, with values of -3.7 mm yr$^{-1}$ and -3.0 mm yr$^{-1}$ per grid-cell over the NH, respectively (Fig. S14). Spatially, the decreased $E_{trans}$ for tree PFTs are mainly distributed in tropical regions and eastern North

America while the decreased grass $E_{subli}$ is located at high latitudes (Fig. S15). Both of these decreases are related to the

variations in surface properties (the decreased $C_d$ and / or decreased $T_{surf}$) in MICT-teb (Fig. 5). As a result of the balance

between the variations of ET and runoff, the SM (0-2 m) in MICT-teb is 9.9 kg m$^{-2}$ (1.8%) wetter than in MICT over the NH

(Fig. 10).

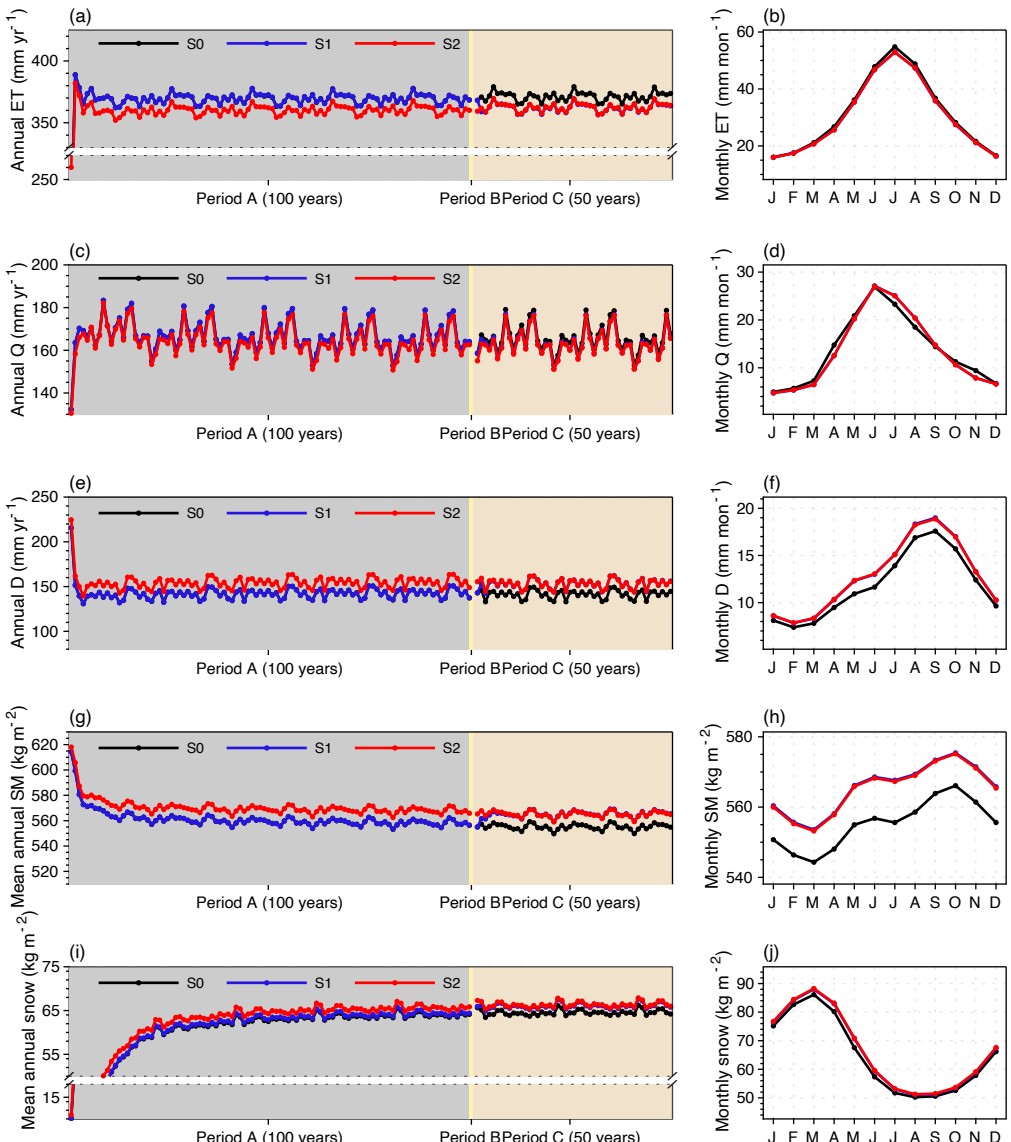

**Figure 10: Time series and seasonal cycle of hydrology variables including evapotranspiration (ET), surface runoff (Q), drainage (D), soil moisture (SM), and snow over the Northern Hemisphere from three simulations.**

### 5.2.3 Carbon

In response to the warmer and wetter soils (Fig. S16), vegetation productivity is significantly enhanced over the NH in MICT-
teb, with a 1.9 PgC yr$^{-1}$ (2.7%) larger GPP and a 0.8 PgC yr$^{-1}$ (2.7%) larger NPP under S2 than S0 (Fig. 11). The enhanced
productivity is primarily driven by tree PFTs across almost all NH regions and grass PFTs at mid- and low latitudes, while the
productivity of peatland grass is somewhat lower due to the cooler soil (Figs. S16 and S17). The warmer soil also accelerates
the heterotrophic respiration rate ($R_h$) for tree and grass PFTs, along with the variation in SM. Compared to S0, S2 has a larger
SOC (+58.5 PgC, 9.0%) for tree PFTs but almost unchanged SOC storage (+22.5 PgC, 2.8%) for grass PFTs. Despite being
the smallest SOC pool (~27%, ~600 PgC) (Hugelius et al., 2014, 2016, 2020; Lindgren et al., 2018) among the three vegetated
soil tiles as a result of the small peatland area (~3% of vegetated land in the NH), the peatland PFT's SOC storage increases
by 85.8 PgC, accounting for more than a half of the total SOC increase from S0 to S2 (Fig. 12). The cooler soil throughout the
entire vertical profile of peat PFT promotes the SOC accumulation by significantly slowing down the soil respiration ($R$ =
0.38, $p < 0.01$), showing a more critical role in regulating the SOC decomposition than SM ($R$ = 0.20, $p < 0.01$). Moreover,
unlike the energy and water processes, the difference in SOC ($\Delta$SOC) over the NH between S2 and S0 is obviously larger than
that between S1 and S0 (~7 PgC), and from a temporal perspective, it exists after period B and then keeps stable until the end
of period C (Fig. 11(g)). This means that the ~170 PgC $\Delta$SOC between S2 and S0 has been accumulated since the peat initiation
and the long-term effects of TEB on soil carbon cannot be neglected.

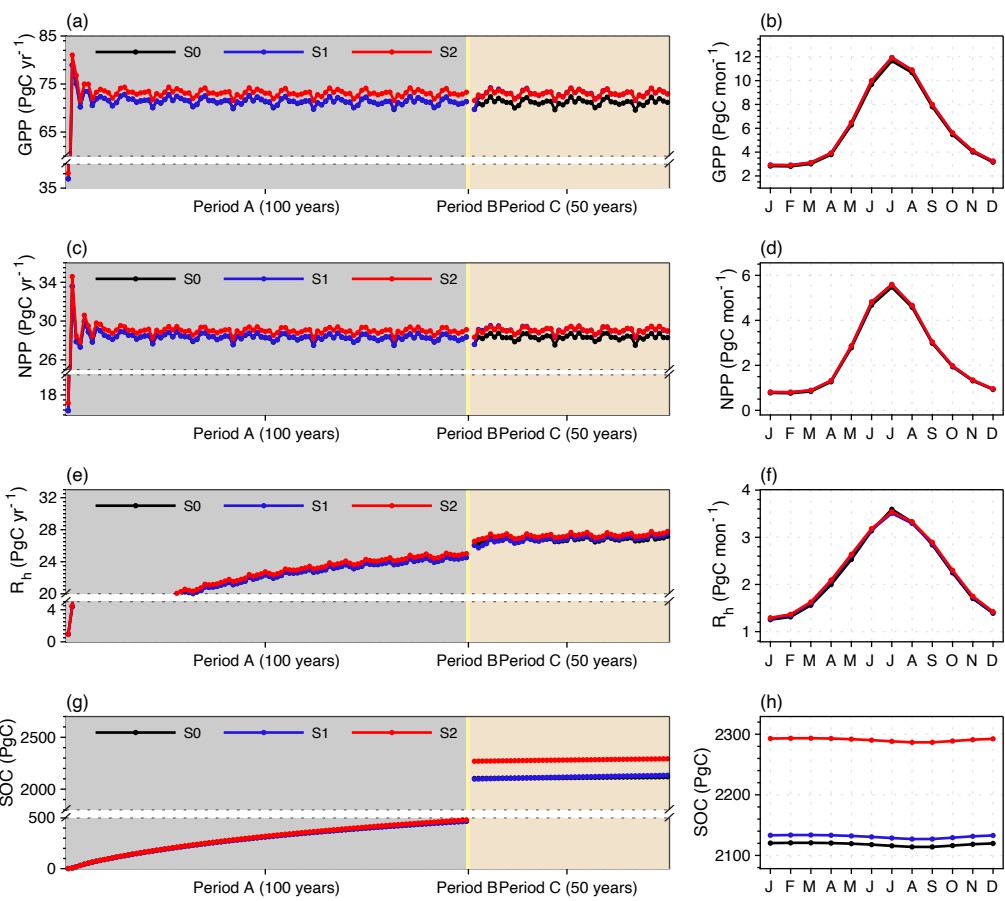


**Figure 11. Time series and seasonal cycle of carbon-related variables including gross primary productivity (GPP), net primary productivity (NPP), heterotrophic respiration (R$_h$), and soil organic carbon (SOC) over the Northern Hemisphere from three simulations. The depth for R$_h$ and SOC is 0-38 m.**

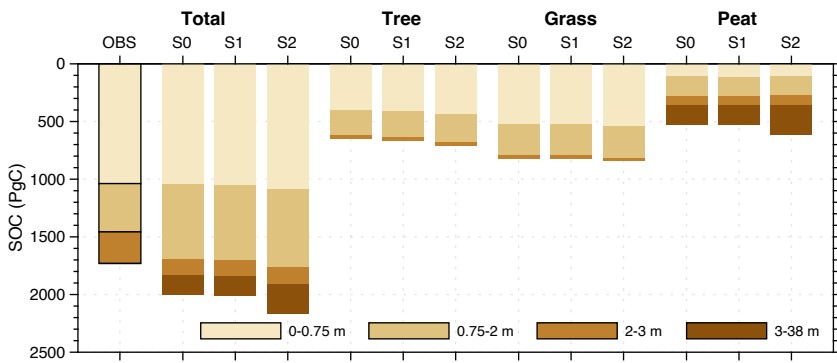


**Figure 12. Vertical composition of soil organic carbon (SOC) for all PFTs and three vegetated soil tiles (tree, grass, and peat) over the Northern Hemisphere from an observation-based SOC map and three simulations. For observed data, we only show the results**

for all PFTs due to the lack of biome information. For simulations, all values are calculated using the average of the last ten years in period C.

## 6 Evaluation and potential application

### 6.1 Evaluation of the simulated energy budgets

To evaluate the simulated surface energy budget, we compare the $T_{surf}$ (surface temperature, mirroring outward longwave radiation and sensible heat flux), albedo (mirroring outward shortwave radiation), and LE (latent flux) from period D (transient simulation, 1901-2020) with satellite-derived land surface temperature (LST), albedo, and LE from MODIS (the Moderate Resolution Imaging Spectroradiometer), respectively. The MODIS LST product is obtained from MOD11C3 Version 6.1 ( https://lpdaac.usgs.gov/products/mod11c3v061/), which records the monthly radiative skin temperature of the land surface at 0.05° × 0.05° spatial resolution, spanning from 2000 to the present (Wan, 2013, 2014). Compared to the MODIS product, MICT and MICT-teb reproduce the spatial pattern of mean annual satellite-derived LST for 2001-2020 (Figs. 13(a)-(c)), but with an overestimation of up to 3 °C in wet regions such as tropical regions, Europe, and eastern North America, as well as an underestimation of up to 3 °C in dry areas and northeastern Asia (Figs. 13(d) and (e)). Regarding the seasonality of $T_{surf}$, the two model versions show an overestimation of summer and autumn LST (by up to 3 °C) but an underestimation of winter LST (by up to 3 °C) in mid-high latitudes, while an up to 3 °C overestimation throughout the year for tropics (Figs. 13(g) and (h)). Including the representation of PFT-specific energy budgets alleviates the LST bias from MICT in some areas such as western North America and northern Europe and in some seasons such as autumn in high latitudes, but at the same time, aggravates the LST bias in some areas and some seasons such as all four seasons in tropical regions (Fig. 13(f)).

The MODIS albedo product is obtained from MCD43C3 Version 6.1 (https://lpdaac.usgs.gov/products/mcd43c3v061/), including the daily black-sky and white-sky albedo at 0.05° × 0.05° spatial resolution. To compare with the albedo from the model, we calculate the blue-sky albedo using the black-sky and white-sky albedo for the shortwave band (0.3-5.0 µm) from MODIS, weighted by the diffuse skylight ratio derived from the direct and total shortwave radiation from the fifth generation ECMWF reanalysis product (ERA5, https://cds.climate.copernicus.eu/cdsapp#!/dataset/reanalysis-era5-single-levels-monthly-means?tab=form) (Hersbach et al., 2023). We found that except a ~0.2 overestimation of winter and spring albedo in northern high latitudes, the simulated albedo from MICT and MICT-teb show a very small bias, no more than 0.04 (Fig. 14). The considerable albedo biases in winter and spring could be related to the bias of simulated leaf area index and snow cover fraction by the model (Li et al., 2016). The MODIS LE product is obtained from MOD16A2GF Version 6.1 (https://lpdaac.usgs.gov/products/mod16a2gfv061/), providing the 8-day LE at 500 m × 500 m spatial resolution. Compared to the MODIS LE, the simulated LE tends to be smaller (-6 - -24 J m$^{-2}$ s$^{-1}$) in most areas except for some arid regions over the NH (Fig. 15). Same as the evaluation for $T_{surf}$, the representation of PFT-specific energy budgets doesn't reduce the albedo / LE biases significantly (Figs. 14(g) and (h), 15(g) and (h)).


There are several reasons that could explain disagreements of surface energy budgets between the MODIS products and the models. On the one hand, 1) the under-representation of some important processes in the model such as the parameterization of ET (LE) and snow insulation, as well as the uncertainties of climate forcing data (Guimberteau et al., 2018; Peng et al., 2016; Domine et al., 2016) and 2) missing data across dry areas and cloudy-weather days in the MODIS products, e.g., a ± 2

°C LST bias compared to ground-truth data found in dry areas (Li et al., 2014; Wan, 2014; Westermann et al., 2012) could partly account for disagreements between the MODIS products and the models. On the other hand, the considerably different land cover maps used by MODIS and the simulations (Fig. S18) and the difference in one specific variable from MODIS and the simulations could contribute to the systematic biases / gaps. For instance, the LST from MODIS reflects more the radiative skin temperature, i.e., canopy temperature, while the canopy energy budget is absent in ORCHIDEE-MICT, which could result

in a higher $T_{surf}$ from the models compared to the MODIS LST in forest ecosystems (Fig. 13) (Gomis-Cebolla et al., 2018).

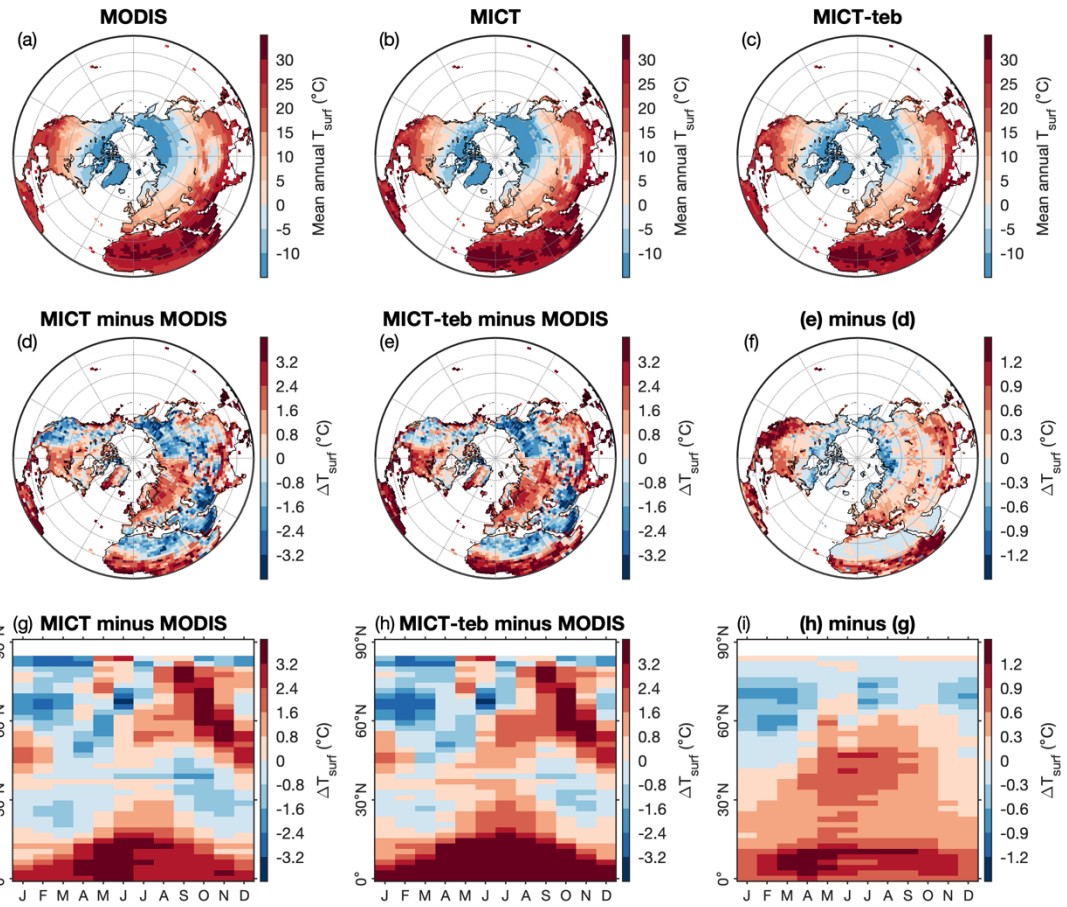

**Figure 13. Evaluation of simulated surface temperature ($T_{surf}$) with land surface temperature (LST) from MODIS. (a)-(c), Spatial pattern of mean annual LST for 2001-2020 from MODIS, MICT, and MICT-teb. (d) and (e), Spatial pattern of the difference in**

mean annual $T_{surf}$ for 2001-2020 between MICT (d) or MICT-teb (e) and MODIS. (f), Spatial pattern of the difference in (e) and (d). (g) and (h), Seasonal cycle of the difference in $T_{surf}$ calculated over each latitude band between MICT (g) or MICT-teb (h) and MODIS. (i), Seasonal cycle of the difference in (g) and (h).

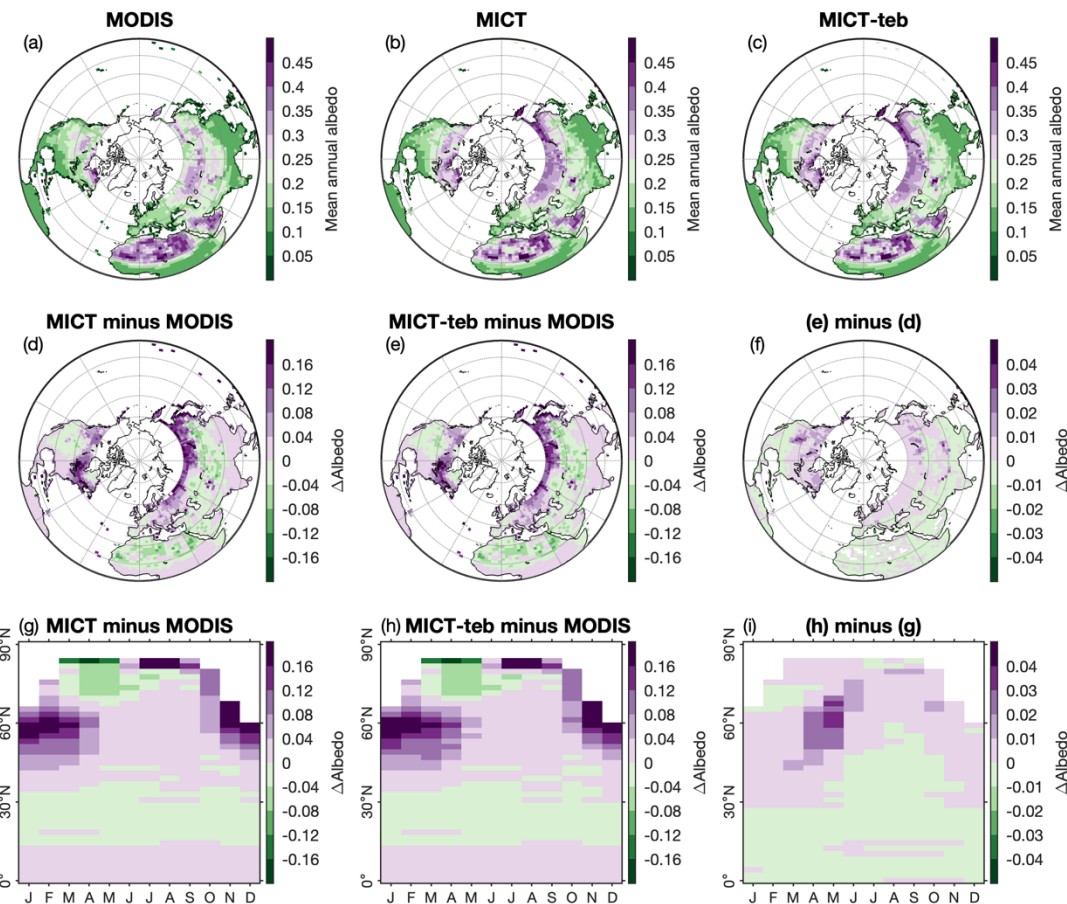

**Figure 14. Same as Figure 13, but for the evaluation of simulated albedo with blue sky albedo from MODIS.**

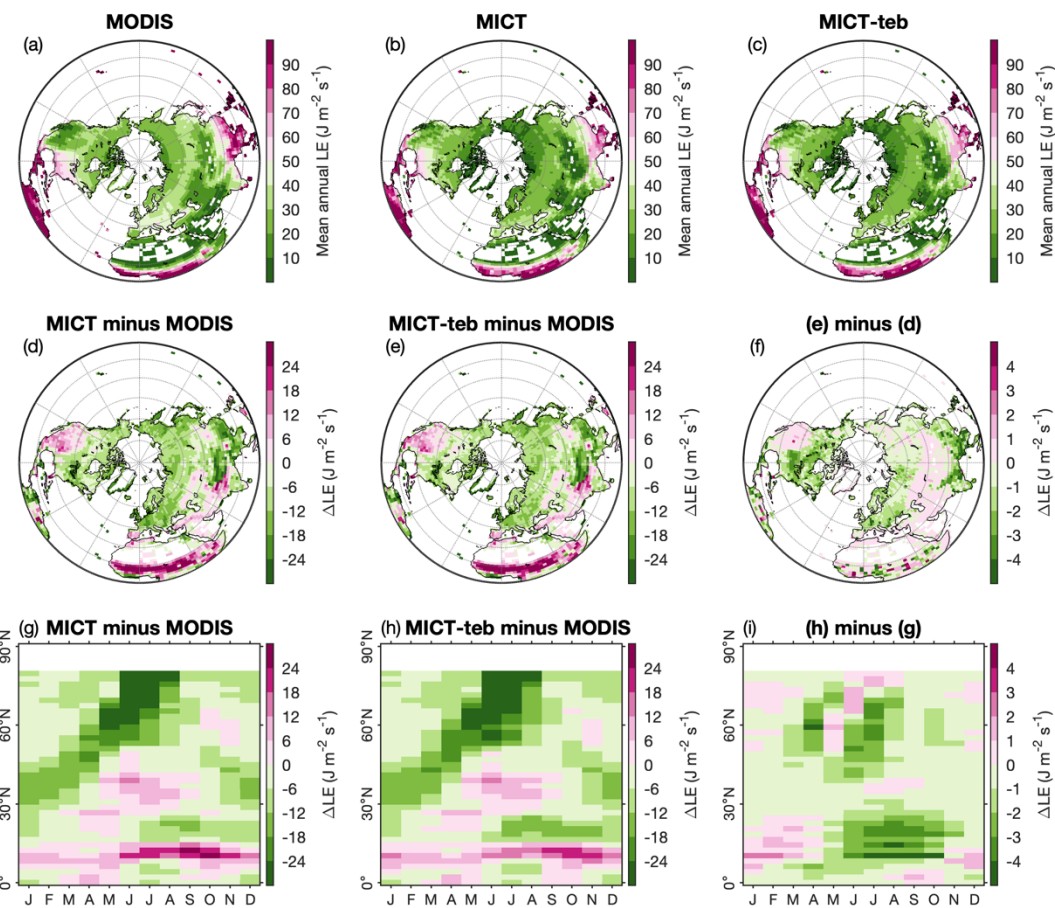

**Figure 15. Same as Figure 14, but for the evaluation of simulated latent flux (LE) with that from MODIS.**

In light of the new feature of MICT-teb to simulate PFT-specific energy budgets, we compare the simulated and satellite-based $T_{surf}$ over grid-cells dominated by four PFT types: bare soil, tree, grass, and peatland from simulations (Fig. 16). Despite the significant disagreements of surface energy budgets between the MODIS products and the models, the model can produce a comparable $T_{surf}$ to MODIS for four PFT types. Also, the model can capture the variations of LST from MODIS well when altering the threshold fractions for grid-cell selection from 50% to 90%. For tree, grass, and peat, due to their extensive

coverage in wet areas, the simulated $T_{surf}$ is 1-3 °C warmer than grid-cell LST from MODIS for all threshold fractions. For bare soil, the underestimation of $T_{surf}$ in desert areas is offset by the overestimation in Greenland (Figs. 13(d) and (e)). The notable biases when using a 90% threshold fraction, a near-complete coverage of one PFT type within one grid cell, suggests the disagreement between the model and satellite products should be attributed to the gap between MODIS and the original version, rather than the separation of PFT-specific energy budgets.


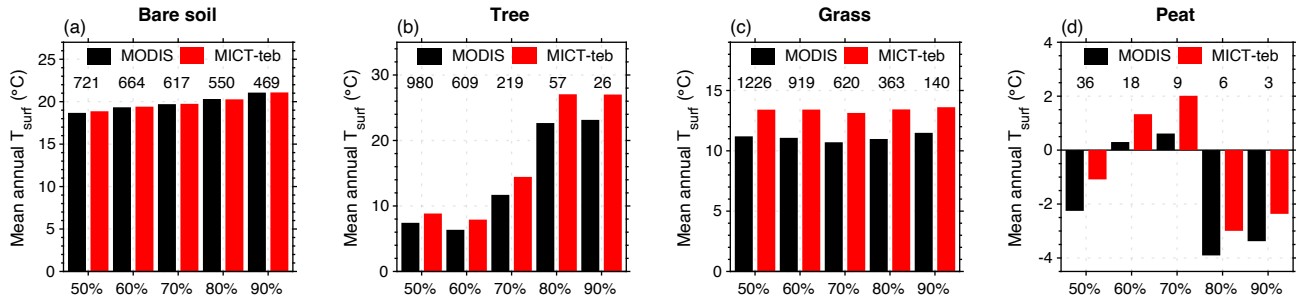

**Figure 16. Evaluation of simulated surface temperature (T$_{surf}$) with land surface temperature (LST) from MODIS by PFT type. (a)-(d), Mean annual T$_{surf}$ for four PFT types (bare soil, tree, grass, and peat) for 2001-2020. The grid-cells for each PFT type are selected with five threshold fractions (50%, 60%, 70%, 80%, 90%), i.e. the minimum fractional coverage of one land cover type in the grid-cell. The numbers of grid-cells for each group of bars are shown.**

For the simulated soil energy budget, we evaluate the T$_{soil}$ at 20 cm with in-situ observations across 268 sites from Russian Meteorological Stations (Sherstiukov, 2012). The period of the site data spans from 1980 to 2000. To avoid the bias resulting from missing values, we exclude 69 sites containing missing values for a specific month in over half of years. As shown in Fig. 17 and Fig. S19, the simulated mean annual T$_{soil}$ at 20 cm from MICT and MICT-teb can well reproduce the spatial gradient of observed values, with a strong and positive correlation (R = 0.94; p < 0.001). However, when it comes to the mean seasonal cycle of T$_{soil}$, a warmer T$_{soil}$ during winter but a cooler T$_{soil}$ during summer are simulated in comparison with observations, especially for sites located in continuous permafrost regions (Fig. 17(c)). Apart from climate-forcing data uncertainties as suggested by Guimberteau et al. (2018), SM, SOC, and snow cover are three factors most likely to regulate the seasonal amplitude of T$_{soil}$. Guimberteau et al. (2018) have demonstrated that the snow insulation is underestimated by MICT when using both GSWP3 or CRUNCEP datasets to force the model. This implies that, from a snow perspective, our simulation should exhibit an amplified seasonal cycle. Conversely, the simulated dampened seasonal amplitude of T$_{soil}$ indicates the potentially crucial roles of SM and SOC. Due to the inclusion of peat PFT in our simulations and the use of simulated SOC rather than prescribed SOC maps to regulate soil thermal properties, the uncertainties in prescribed peatland maps and simulated SOC could propagate the uncertainties in T$_{soil}$. Moreover, we extract LST in 2000 across these sites from MODIS data and find a weak correlation between the bias in T$_{surf}$ against MODIS and the bias in T$_{soil}$ against site data (R = 0.19, p < 0.01 for all sites; R = -0.01, p > 0.05 for continuous permafrost sites; R = 0.03, p > 0.05 for other sites), suggesting the potential uncertainties in observed data from different sources.

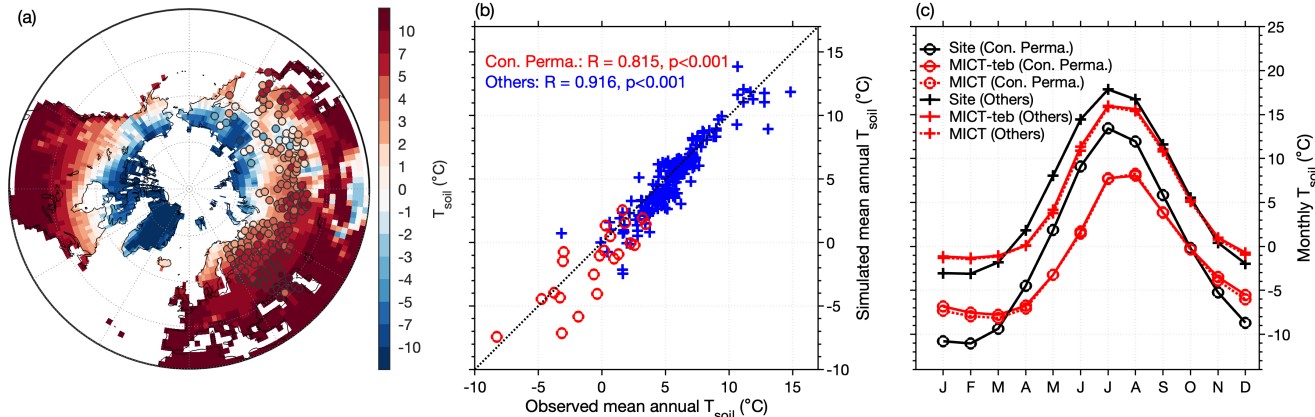

**Figure 17. Evaluation of simulated soil temperature ($T_{soil}$) at 20 cm with site observations. (a),** Spatial patterns of mean annual $T_{soil}$ at 20 cm during the period 1980-2000 from the simulation of MICT-teb, with the values of 199 sites shown as color filled circles. **(b),** Simulated (from MICT-teb) versus observed mean annual $T_{soil}$ at 20 cm across all sites. **(c),** Mean seasonal cycle of site-averaged $T_{soil}$ at 20 cm from site observations, MICT-teb, and MICT. The sites in (b) and (c) are divided into those located in continuous permafrost (Con. Perma.) regions (22 sites, circle markers) and others (177 sites, markers of plus sign) according to the permafrost map of Brown (2002).

As for the impacts of tiling energy budgets on $T_{surf}$, our simulations suggest that the variation in albedo plays a dominant role at high latitudes, whereas the variation in $H_{rough}$ is more significant at low latitudes (Table 2). This result is consistent with numerous studies on the biophysical feedbacks of land cover change, as the use of PFT-specific $H_{rough}$ and albedo for tree PFTs can be seen as an analogy to reforestation or afforestation, while the use of PFT-specific $H_{rough}$ and albedo for grass PFTs parallels the case of deforestation or forest degradation. Flux tower measurements, satellites, and climate models have revealed that tropical tree planting mitigates warming through evaporative cooling while the low albedo of new boreal forests is a positive climate forcing (Betts, 2000; Bonan, 2008; Peng et al., 2014; Su et al., 2023). In contrast, the conversion of forests to grasslands, or forest degradation shows a warming effect in tropical regions but a cooling effect in boreal regions (Lawrence and Vandecar, 2015; Ramdane Alkama and Alessandro Cescatti, 2016; Li et al., 2022; Zhu et al., 2023). This consistency confirms the modifications we made in the new MICT version.

## 6.2 Improvements for permafrost simulations

Given the crucial role of $T_{soil}$ in permafrost simulations, we compare the simulated permafrost extent from MICT and MICT-teb with two independent permafrost datasets from Brown et al. (2002) and Obu et al. (2019), hereafter named Brown2002 and Obu2019, respectively (Fig. 18). Brown's map is compiled based on national / regional maps and empirical knowledge, categorizing permafrost into four classes: continuous permafrost (permafrost fraction > 0.9), discontinuous permafrost (0.5 ~ 0.9), sporadic permafrost (0.1 ~ 0.5), and isolated patches (0 ~ 0.1) (Brown, 2002). Obu's map is generated by using a

temperature model to simulate soil thermal regimes in 3D at 300 m × 300 m spatial resolution (Obu et al., 2019). The Obu2019 data provides the absolute fraction of the landscape affected by permafrost when aggregated to a coarser grid-cell (Obu et al., 2019). The simulated permafrost areas are identified following Guimberteau et al. (2018) with two definitions: 1) active layer thickness (ALT) less than 3 m; or 2) $T_{soil}$ of any soil layer remains below 0 °C for at least two years. Since the simulated permafrost areas are very similar between the two definitions, we only show the results using the ALT definition in the manuscript while the results using the $T_{soil}$ definition in supplementary (Fig. S20).

Overall, our models can capture the spatial pattern of continuous permafrost from the two independent datasets. However, the simulated total area of continuous permafrost from MICT, after excluding Greenland (18.5 Mkm$^2$) is 7.6 and 6.9 Mkm$^2$ larger than that from Brown2002 and Obu2019, primarily due to the overestimation of continuous permafrost in mid-high latitudes of Asia and the Tibetan Plateau. Given that the grid-cell mean $T_{soil}$ in MICT-teb is ~ 0.5 °C warmer over the NH than in MICT (Fig. 8), the total area of continuous permafrost simulated by MICT-teb (15.2 Mkm$^2$) reduces by 3.3 Mkm$^2$ than that in MICT. Limited by the grid-cell averaged energy budgets, MICT can't simulate non-continuous permafrost, that is a grid-cell in MICT is either a 100% permafrost or a 100% non-permafrost. While in MICT-teb, the separation of PFT-specific $T_{soil}$ allows the existence of non-continuous permafrost (Fig. 18(e)). The sum of all four permafrost areas (including three discontinuous classes) in MICT-teb is 18.3 Mkm$^2$ (continuous 15.2 Mkm$^2$, non-continuous 3.1 Mkm$^2$), which is comparable to 16.9 Mkm$^2$ in Obu2019 (continuous 11.6 Mkm$^2$, non-continuous 5.3 Mkm$^2$).

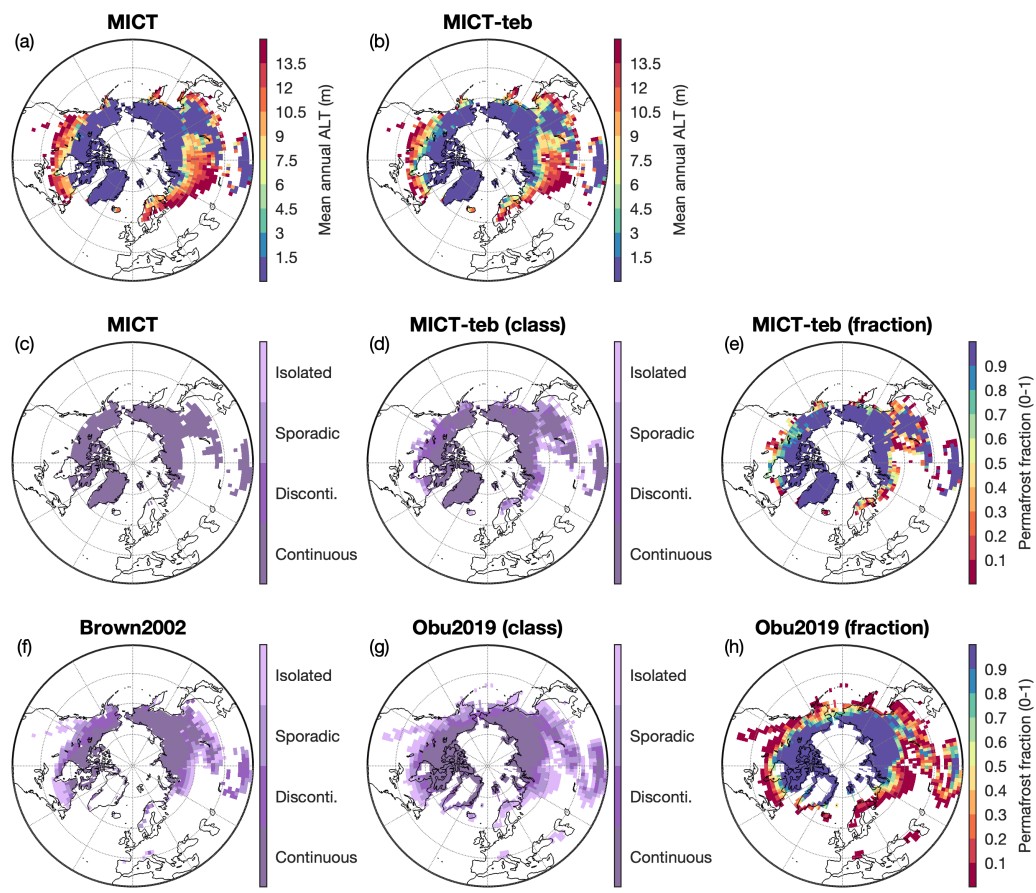

**Figure 18. Evaluation of simulated permafrost areas with independent permafrost datasets. (a)-(b), Spatial patterns of active layer thickness (ALT) simulated by MICT and MICT-teb. (c)-(e), Spatial patterns of permafrost areas simulated by MICT and MICT-teb according to the definition of ALT. (f)-(h), Spatial patterns of permafrost areas from Brown2002, Obu2019. The permafrost areas in Figs. (c), (d), (f), and (g) are shown following four permafrost classes from Brown2002, while the permafrost areas in Figs. (e) and (h) are shown with absolute fraction.**

## 6.3 Remarks on the tiling land surface scheme

The tiling work in this study was initiated because of the planned introduction of new arctic landforms for permafrost regions into ORCHIDEE-MICT that requires independent energy budgets, carbon and water cycles. The decision to tile by PFT, rather than other units, was determined based on the current model structure. In the new version, additional variables with a new PFT dimension were introduced only for the energy module, rather than all modules, resulting in a 15-20% slower run time compared to the initial version. Recently, several other model groups have also been working on the tiling and the evaluation of its impacts on existing and new processes, such as JULES (Rumbold et al., 2023) and CLASSIC (Melton et al., 2017). The implementation of tiling in different land surface models can be compared to inspire other groups planning to represent sub-

grid heterogeneity of energy budgets in their models. Moreover, for potential users of our new version, it is worth reminding them to carefully consider when and where to apply tiling in their studies to optimize research objectives and computational 650  costs.

## 7 Conclusion

This study describes the new representation of tiling energy budgets in the ORCHIDEE-MICT land surface model and investigates its short and long-term impacts on energy, hydrology, and carbon processes. Instead of using grid-cell mean 655  surface properties like roughness height and albedo, the PFT-specific values are employed for each vegetation type and all of the associated energy, hydrology, and carbon processes are modified in the model. Compared to the original version, the separation of PFT-specific energy budgets results in warmer surface and soil temperatures, higher soil moisture, and increased soil organic carbon storage across the Northern Hemisphere. Evaluation with satellite products and site measurements suggests that the new version can reproduce the spatial distributions and seasonal patterns of surface and soil temperature. However, 660  notable positive or negative biases are observed at some regions due to remaining weaknesses of the original ORCHIDEE-MICT model, such as the uncertainties in simulating soil moisture and soil organic carbon, as well as uncertainties in the prescribed peatland map. A notable advancement in the new version is the improved simulation of permafrost extent by accounting for the presence of discontinuous permafrost, which will facilitate various permafrost-related studies based on the model in the future.

665

**Code availability.** The ORCHIDEE-MICT-teb model (r8205) code used in this study is open-source and distributed under the CeCILL (CEA CNRS INRIA Logiciel Libre) license. It is deposited at https://forge.ipsl.jussieu.fr/orchidee/wiki/GroupActivities/CodeAvalaibilityPublication/ORCHIDEE-MICT-teb and archived at https://doi.org/10.14768/0954a0e9-6a7a-4006-803e-4db36ef2db88, with guidance to install and run the model at 670  https://forge.ipsl.jussieu.fr/orchidee/wiki/Documentation/UserGuide. Codes to process data, generate all results, and produce all figures are archived at https://doi.org/10.5281/zenodo.10014533 (Xi, 2023).

**Author contributions.** PC conceived the project. YX implemented tiling energy budgets into ORCHIDEE-MICT. CQ, YZ, DZ, and SP provided general scientific guidance to improve the research and interpret results. YX performed the simulations, 675  did the analysis, and wrote the paper. All authors contributed to commenting and writing on the draft manuscript.

**Competing interests.** The contact author has declared that none of the authors has any competing interests.

**Acknowledgements.** This research was supported by the European Union's Horizon 2020 research and innovation programmes under grant agreement No 101003687 (PROVIDE). PC acknowledges support from the CALIPSO (Carbon Loss in Plant Soils and Oceans) project, funded through the generosity of Eric and Wendy Schmidt by recommendation of the Schmidt Futures program. DZ acknowledges support from National Natural Science Foundation of China (No. 42101090).

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
