# Peer review of "Assessment of a tiling energy budget approach in a land surface model, ORCHIDEE-MICT (r8205)"

_Geoscientific Model Development, 2023_

## Author Comment (AC1)

**Response to the reviewers**

**General comments**

*Xi et al. add an expanded tiling approach to ORCHIDEE-MICT. This new approach, termed MICT-teb allows for per-PFT tiling of the energy budget with effects cascading to the hydrology and carbon cycle. The paper reads OK but has issues with clarity in some sections, which I note below. The actual approach is relatively straight forward and present a comprehensive overview of how MICT changed with tiling. Many groups have done this in their models and these results are similar to others, but it is interesting to see how it was implemented in ORCHIDEE and the impacts. One thing that I missed in the paper was a consideration of what actually makes the most sense to tile. Here they tiled by PFT, as others have done before, but the decision seemed to just be a default option rather than one carefully considered. Based upon my experience, I am not sure if tiling by PFT makes the most sense. In some landscapes the PFTs are truly distinct (e.g. peatlands vs. nearby uplands), but in others they likely shouldn't be separated (e.g. savannah landscapes). I would like to see some more discussion about their choice to tile based upon PFT and then at the end a discussion of whether that choice was a good one. Within our group it has been a learning process of when to tile and when it doesn't seem worth the complexity (e.g. we have tried by PFT - Shrestha et al. 2016, soil texture - Melton et al. 2017, and by disturbance - Curasi et al. 2023). I personally seem to have landed on peatland vs. upland and by disturbance requiring tiling, but I would be quite interested to hear of the authors' finding here. Otherwise, I have many small comments but don't foresee a problem with them being addressed so can recommend publication after acceptable revisions.*

**[Response]** Thank you so much for your time in reviewing our manuscript and for your positive feedback on our work. We also appreciate your sharing of the tiling experiences with your model. The reasons why we tiled by PFT are as follows:

1) The purpose behind initiating this tiling work is to introduce new landforms for permafrost regions in ORCHIDEE-MICT, such as peatland vs nearby uplands, and landforms with or without ground ice. The landforms with and without ground ice can be described by tiles that must have different energy budgets, water and carbon cycles. Therefore, a model with the ability to simulate distinct energy budgets within one grid cell is an essential prerequisite for most of our planned developments for permafrost regions.

2) We chose to tile by PFT rather than other units because tiling by PFT is the easiest way for us to have distinct energy budgets within one grid cell, based on the existing model structure. In the current ORCHIDEE-MICT, the carbon cycle is conducted for each PFT and the water cycle is conducted for each soil tile. There are four soil tiles including bare soil PFT, all tree PFTs, all grass and crop PFTs, and one peatland PFT in ORCHIDEE-MICT, based on the simplified assumption that different tree (or grass) PFTs have similar behaviors in the water cycle. Unlike the water and carbon cycles, the energy budgets from the surface to the soil are conducted for the entire grid-cell with the grid-cell mean surface, or snow, or soil properties, except for the latent heat flux, i.e. evapotranspiration due to the significantly

different characteristics in the carbon module among different PFTs. Since the energy budgets are closely related to water and carbon dynamics, some variables in energy budgets of the current model include the PFT dimension to make the connection between energy and water or carbon much easier technically. However, the values for all PFTs for this dimension are the same. In this case, it's not very complicated to use specific values of surface, snow, or soil properties for each PFT to separate the calculation of energy budgets within one grid cell, as well as refine the PFT-specific connection between energy and water or carbon in the model. Moreover, the separation was done only for the energy module, rather than all modules, resulting in a 15-20% slower run time than the initial version. The extra computational cost is quite acceptable.

On the basis of the version with PFT-specific energy budgets, we plan to introduce new landscape elements (or landforms for Arctic) as new PFTs in ORCHIDEE in the future and provide specific attributes for its carbon cycle, water cycle and energy budgets. In fact, the introduction of one peatland PFT has served as such an example (Qiu et al., 2018, 2019, GMD). Qiu et al. defined specific attributes and processes in water and carbon cycles for the peatland PFT, such a porosity and carbon content resulting in a moister soil and a deeper soil organic carbon for the peatland PFT compared to non-peatland PFTs. The significant difference in surface and soil temperature between peatland PFT and non-peatland PFTs has been investigated further in this manuscript when using the new version with PFT-specific energy budgets.

Regarding the question of for which ecosystems separate tiles are needed, we totally agree with the reviewer that it's indeed a crucial question to consider. In areas with mixed trees and grasses having small-scale interactions between them, such as savannah landscapes, or for mixed bare soils and variable plant cover, there is no necessity to implement tiling energy budgets to increase the complexity. Our work only enables ORCHIDEE-MICT conduct the tiling energy budgets; it does not suggest applying tiling energy budgets everywhere. And our work is suitable for the Arctic where hydrologically distinct landforms or landscape elements can be described reasonably by tiles.

Thank you again for bringing these questions into our attention and making us hear concerns from other models. Following your suggestions, we have added a new paragraph to the discussion, covering the reasons behind our choice to tile by PFT, the extra computation cost of the new version, and the reminder on when and where to tile for the potential users of our version (copied as below, L639-649). Then please find detailed point-by-point responses to each of your small comments.

"**6.3 Remarks on the tiling land surface scheme**

**The tiling work in this study was initiated because of the planned introduction of new arctic landforms for permafrost regions into ORCHIDEE-MICT that requires independent energy budgets, carbon and water cycles. The decision to tile by PFT, rather than other units, was**

determined based on the current model structure. In the new version, additional variables with a new PFT dimension were introduced only for the energy module, rather than all modules, resulting in a 15-20% slower run time compared to the initial version. Recently, several other model groups have also been working on the tiling and the evaluation of its impacts on existing and new processes, such as JULES (Rumbold et al., 2023) and CLASS (Melton et al., 2017). The implementation of tiling in different land surface models can be compared to inspire other groups planning to represent sub-grid heterogeneity of energy budgets in their models. Moreover, for potential users of our new version, it is worth reminding them to carefully consider when and where to apply tiling in their studies to optimize research objectives and computational costs."

**Smaller comments**

*Line 48: Cooling effect already implies it is a reduction so should just be 2.5C, not -2.5C.*
**[Response]** The minus sign has been removed as suggested.

*L 55: The sentence should be rewritten for clarity.*
**[Response]** Together with comments from the other reviewer, we have rewritten the sentence as "**Through the comparison between the "mosaic" and the "composite" LSSs, the CLASS model reported a less than 5% difference in the primary energy fluxes but an up to 46% difference in carbon fluxes and carbon pool size at site level (Li and Arora, 2012), as well as a 19% higher terrestrial carbon sink for 1959-2005 in the "mosaic" simulation (Melton and Arora, 2014).**" (L59-62).

*Fig 1. I found this to be relatively hard to understand. First, the caption includes a duplicate jibberish caption label. For the figure itself, I am not sure what the red rectangles mean. The MICT-teb has only one but the MICT has two(?). I think this could better be redrawn for clarity. There are other model schematics in the literature that the authors could look to for inspiration.*
**[Response]** First, thanks for the kind reminder and we have removed the duplicate jibberrish caption label. Then for the figure itself, we expected to distinguish grid-cell mean and PFT-specific calculations with red and blue arrows / rectangles, respectively in the initial manuscript. Following the reviewer's suggestion, we kept one red rectangle both for MICT and MICT-teb, indicating one model grid cell in the revised version (copied here as Figure R1). For MICT, the red rectangle is filled with dots evenly, together with the red arrows for heat fluxes, indicating that the energy budgets are conducted for the entire grid cell. For MICT-teb, all heat fluxes are represented using blue arrows, indicating the energy budgets are conducted for each PFT. We have also added one sentence "**The red and blue arrows distinct the grid-cell mean and PFT-specific calculations.**" in the caption to make it clearer to readers.

[Figure]

Figure R1 (Figure 1). Schematic representation of energy budgets at the surface, snow layers, and soil layers in one grid cell of ORCHIDEE-MICT (MICT) (a) and the new tiling energy budget version (MICT-teb) (b). $SW_{in}$, $SW_{out}$, $LW_{in}$, $LW_{out}$, H, and, LE represent incoming ShortWave radiation, outward ShortWave radiation, incoming LongWave radiation, outward LongWave radiation, sensible heat flux, latent heat flux, respectively. PFT indicates Plant Function Type. There are 3 layers for snow, and 32 layers for soil for each PFT in the model. In MICT, $SW_{in}$, $SW_{out}$, $LW_{in}$, $LW_{out}$, H, and heat fluxes in snow and soil layers are calculated as grid-cell mean but LE is calculated for each PFT, while in MICT-teb, all of the heat fluxes are calculated for each PFT. **The red and blue arrows distinguish the grid-cell mean and PFT-specific calculation.**

*Line 111: Reference for the <5% number?*

**[Response]** The reference (Georg et al., 2016) has been added for this number.

*L 123: specific heat 'capacity' of dry air*

**[Response]** The "capacity" has been added.

*Table S1 - are your layer thicknesses really so thin? In our model we find stability issues if the layer thickness goes below about 10 cm. The first layer here seems to be 1 mm thick(?!). Also to be clear - this shows depth to the bottom of the layer or layer thickness?*

**[Response]** Yes, the layer thickness is very thin (less than 10 cm) for the first several layers in ORCHIDEE-MICT, but we didn't meet any stability issues for the heat transfer. The depth shown here is the depth to the bottom of the layer. We have added an explanation of the depth in the caption following the reviewer's suggestion.

*line 184: missing 'taken'*

**[Response]** The 'taken' has been added.

*eqn 5: why heat capacity now lower case, was upper case earlier.*

**[Response]** Following the reviewer's suggestion, we have revised the heat capacity for dry air ($C_p$) as $c_p$ in eqn (3) to keep a consistent case for heat capacity throughout the manuscript.

*Table S2: Minimum snow albedo after aging can be incredibly low (0.14). Even the oldest, melting snow should have albedo around 0.5 as far as I have ever seen. Can these low values be justified?*

**[Response]** Thanks for the question. In Table S2 (Table S3 in the revised version), the minimum snow albedo after aging ($\alpha_{snow\_min}$) for PFT 4-16 is 0.14 or 0.18, and the fresh snow albedo ($\alpha_{snow\_min} + k$) varies from 0.22 to 0.74. The minimum snow albedo or even the fresh snow albedo of some PFTs is lower than 0.5 as the reviewer thought, because it is the case when the snow is shaded by leaves. The shading effect of leaves could reduce the snow albedo greatly. When there is no leaf cover, i.e. for bare soil (PFT1), the minimum snow albedo after aging increases to 0.35 and the fresh snow albedo increases to 0.8, which is more consistent with that pointed out by the reviewer.

Since the PFT cover fraction in ORCHIDEE includes two parts, one is leaf-covered area and the other is no-leaf-covered area, i.e. bare soil (see more details about the splitting of leaf-covered area and bare soil for one PFT in the following response), the minimum snow albedo after aging for one PFT varies from the leaf-covered $\alpha_{snow\_min}$ to no-leaf-covered $\alpha_{snow\_min}$. Figure R2 showed the monthly snow albedo for PFT9 (boreal needleleaf summergreen tree) in a randomly selected year from our simulations. The values of the snow albedo are close to the range suggested by the reviewer, increasing from no more than 0.4 in summer to ~0.8 in winter.

[Figure]

Figure R2. Spatial pattern of monthly snow albedo for PFT9 (boreal needleleaf evergreen tree) from our simulations.

Additionally, all the values of snow albedo parameters we used here are defaulted by ORCHIDEE-MICT. They were from the PhD thesis of Sophie Najm Chalita (1992) and the corresponding journal article (Chalita et al., 1994), where the parameterization of the snow albedo has been evaluated with site measurements (Robinson and Kukla 1984). In the revised version, we have added the explanation of snow albedo parameters for vegetated PFTs for Table S3 (copied as below) and the corresponding references to help understanding.

"**Note that the two snow albedo parameters for PFT 2-16 in the model consider the shading effects of leaves, therefore with a smaller value than those on bare soil.**"

*Fig S1 - where is the soil albedo from MODIS coming from? This is not straight-forward to produce so assumedly this is from some other source?*

**[Response]** The soil albedo map was produced from ORCHIDEE simulations for the CMIP6 project. We have added the reference (Lurton et al., 2020) in the revised version.

*Fig S2 - this is hard to compare between b and c. The different PFTs are not labelled in a manner that would make comparison simple. As it is now, I can't really tell what the differences are except that the*

*model's SOC drops too quickly with depth. It would be nice if this figure was improved to allow a proper comparison.*

**[Response]** Thanks for the suggestion. We have improved this figure by using the same names and colors for land cover types from the model and observation data (copied here as Figure R3). However, it's still a little hard to compare since the land cover types from the model and observation data are not completely the same. Overall, the simulated soil organic carbon is comparative to the observation data for several common land cover types including DBF, ENF, DNF, and Graminoid / C3G, but the model underestimates the maximum depth of soil organic carbon compared to the observation data except for peatland PFT over the 16 sites.

[Figure]

Figure R3 (Figure S2). Comparison of vertical SOC profile (0-3 m) across different land cover types between site-level data from Palmtag et al., (2022) and simulation from ORCHIDEE-MICT. (a), Spatial distribution of 16 sites used in Palmtag et al., (2022). (b), The vertical profile of mean SOC density across all sites for 10 land cover types (DBF: deciduous broadleaf forest; ENF: evergreen needleleaf forest; DNF: deciduous needleleaf forest; Shrub: shrub tundra; Graminoid: graminoid / forb tundra; PerWet: permafrost wetlands; NPWet: non-permafrost wetland; Barren: barren; YeTundra: yedoma tundra; Yeforest: yedoma forest) from Palmtag et al., (2022). (c), The vertical profile of mean SOC density across all sites for 8 PFTs (EBF: evergreen broadleaf forest; ENF: evergreen needleleaf forest; DBF: deciduous broadleaf forest; DNF: deciduous needleleaf forest; C3C: C3 crop; C4C: C4 crop; C3G: C3 grass; Peat C3G: peatland C3 grass) from ORCHIDEE-MICT. The PFTs without cover fraction and bare soil (without organic carbon) from the simulations are not shown here.

*Fig S3 - the model and ref dataset appear to be on different grids. What are we supposed to make of the inland white cells? Does the model blow up in those locations or ? It would be good to see a difference plot to make it clear where the biases are, against at least this one ref dataset.*

**[Response]** Both the model and the reference dataset were shown at the same spatial resolution (2° · 2°) but with different land-ocean masks in the initial manuscript. The inland white cells were not shown because the soil organic carbon is zero. We have revised this figure by using the same land-ocean mask from the model and showing ocean grid cells in white. Besides, we have added figure (c), the difference between simulated and observation-based soil organic carbon as the reviewer suggested (copied here as Figure R4).

[Figure]

Figure R4 (Figure S3). Comparison of spatial patterns of SOC for 0-3 m from the simulation from ORCHIDEE-MICT and observation-based data as used in Zhu et al. (2019). (a) ORCHIDEE-MICT; (b) the observation-based data; **(c) the difference between ORCHIDEE-MICT and the observation-based data.**

*L 265: provide ref for CRU-JRA*

**[Response]** The CRU-JRA dataset was obtained from the project of Global Carbon Budget 2022. The corresponding reference (Friedlingstein et al., 2022) has been added in the revised version.

*L 266: Provide proper ref for land cover. Yes it is used by TRENDY but it comes from somewhere else. It is a lot of work to create these datasets so the least we can do is properly cite them.*

**[Response]** The reference (Lurton et al., 2020) has been added for the land cover maps in the revised version.

*I suggest you choose either mteb or teb. The model name uses teb but the text uses mteb. It will be more clear if you just use one.*

**[Response]** Thanks for the suggestion. We have used the "teb" throughout the revised manuscript.

*L301 - didn't you say earlier in the paper that albedo was insensitive to soil moisture?*

**[Response]** We earlier mentioned that the albedo of bare soil is insensitive to soil moisture, emphasizing that the bare soil albedo prescribed with the MODIS product is decoupled with simulated soil moisture and therefore without temporal variation (L135-137). While in this context, the mention of soil moisture is intended to explain the spatial variation of bare soil albedo.

*Fig 3: Is the peatland parameterization also a tropical one in addition to a boreal representation?*

**[Response]** Since there's only one peatland PFT in the simulation of this study, the parameterization of tropical peatlands is identical to that of boreal peatlands.

*Line 306: How does this work 'We note that the cover fraction of a PFT in the model includes both valid vegetation and bare soil.' - wouldn't you want the PFT cover fraction to be only the PFT cover and bare soil fraction be accounted for separately? Or is this meant to be talking about the soil below the canopy? If so, I suggest you use different, more clear, terminology.*

**[Response]** The PFT cover fraction in ORCHIDEE-MICT includes not only the canopy, but also bare soil if we call it follow the ORCHIDEE technical documents. The bare soil here is NOT the soil below the canopy, but the area not covered by leaves. It should make sense because the soil may not be 100% covered by leaves in a forest, or a grassland, or a cropland in reality. The leaf-covered area in the model is determined using a function of leaf area index (LAI):

$$veget = veget\_max \times (1 - e^{(-LAI)}) \qquad (R1)$$

where veget_max is the maximum PFT cover fraction, veget is the instantaneous leaf-covered area. Figure R5 shows the change of leaf-covered area over LAI taking three fractions of PFT cover fraction (0.1, 0.3 and 0.5) as examples.

[Figure]

Figure R5. Change of leaf-covered area vs LAI for three fractions of PFT cover fraction (0.1, 0.3 and 0.5) following eqn (R1).

In ORCHIDEE-MICT, the canopy processes such as transpiration in the water cycle and photosynthesis in the carbon cycle can only happen over the leaf-covered area, while the no-leaf-covered area is treated same as PFT1 (100% coverage of bare soil) e.g., when calculating snow albedo for one PFT mentioned earlier. That is why we call it bare soil. We have revised the descriptions in the revised version as "**We note that the cover fraction of a PFT in the model includes both the leaf-covered area (the canopy) and the no-leaf-covered area (the soil), depending on a function of leaf area index. The albedo of a PFT is calculated as the area-weighted sum of the albedo of leaves and the albedo of soil within this PFT.**" (L327-329).

*L 312: So roughness height is not static? Table S4 makes it seem like it is static per PFT. This is confusing.*

**[Response]** The roughness height is static for each PFT. This sentence just states the general relationship between surface drag coefficients ($C_d$) and roughness height ($H_{rough}$), which is used to explain the difference of $C_d$ between one PFT in MICT-teb and the grid-cell mean in MICT from the perspective of $H_{rough}$.

*Fig 4 + 5: I think I know what this shows for the MICT-teb simulation, e.g. bare soil is the values over the bare soil tiles, but I am less certain what that is compared to for the MICT simulation. Looking at Fig 1 makes me think that all panels of this figure will be comparing a MICT-teb tile against the same values from MICT. Is that right? If so, then naturally the differences will be large. The most interesting changes then are the grid-cell mean row. I would be tempted to put the rest in the supplement since the*

*comparison is not quite clear as it is for the grid-cell mean. And then add a second row showing the relative difference (or a second scale as done in Fig 3). Have you also looked at the NH totals for these quantities (where applicable)? This might get at the problem with tiling, in my experience at least, where is it most valuable/needed? It adds a lot of complexity so should add some real benefit. I am wondering about something like Fig S14 but for the whole NH, not just a few cells.*

**[Response]** In Figs. 4 and 5, we showed the comparison for the grid-cell mean between MICT-teb and MICT in first row, and then the comparison between a MICT-teb tile and the grid-cell mean from MICT. So, yes, all panels in Figs. 4 and 5 were using the similar values of the grid-cell mean from MICT. There are several reasons to show the comparison for a MICT-teb tile. On the one hand, the difference in energy budgets between a MICT-teb tile against the grid-cell mean in MICT is easier to understand for readers than the difference of the grid-cell mean between the two versions, due to the clearer direction when comparing surface properties between a tile and the grid-cell average. It would be a better strategy to make readers follow the modifications we've done in the new version and then the consequences after modifications by showing the differences for a tile. On the other hand, the changes in energy budgets of each tile are most directly used to explain the changes in the following carbon cycle and water cycle in the manuscript, because the carbon cycle and water cycle are computed for each tile, rather than the grid-cell mean in the model. Therefore, we still kept the rows for each tile in the two figures. The results for the whole NH and their temporal variations can be found in Figs. 9-11.

*Table 2: Have these effect sizes been checked statistically? It would be good to know which are more significant. Esp if you are assigning an effect size, when you are comparing two things of differnt units (e.g. albedo change and SWout change). I see this is done in Table S5, why not here?*

**[Response]** Table 2 gives a qualitative summary of the change in surface temperature and the changes in surface properties including albedo and roughness height. As explained in our last response, we hope the summary can help readers understand the modifications we've done in the new version i.e., the changes in surface properties and the consequences after modifications i.e., the change in surface temperature. The quantitative difference in these variables can be found in Figs. 4-5 for grid scale and in Figs. 9-11 for the whole NH.

*Fig 6 has a lot of info but similarly to my comment about Fig 4 + 5, an important change is the grid-cell mean, which is not shown here.*

**[Response]** Similar to Figs. 4-5, this figure and Fig. S6 help readers understand the relationship between the changes in surface heat fluxes and the changes in surface properties. In the two figures, the larger differences in surface energy fluxes between one tile of MICT-teb and the grid-cell mean from MICT are found when the cover fraction of one PFT is smaller in the grid-cell. The PFT cover fraction is not applicable for the grid-cell mean (the cover fraction is always 1 for a grid-cell), so the grid-cell mean comparison wasn't shown here.

*L 414: Assumedly the quick equilibration is due to defined vegetation heights? If the vegetation could grow according to conditions, that should take longer than 2-3 years to find a new equilibrium.*

**[Response]** Yes. The vegetation growth is not dynamic in our simulations, resulting in the quick equilibration here.

*L 424: But surely the subtle effects are when looking at grid cell mean values and not, e.g. the peat tile specifically? I would expect the peat tile to be much different (as shown in Fig 8). Perhaps tighten up the language here to be more specific.*

**[Response]** Thanks for the suggestion. We remind the reviewer that the differences shown in Sect. 5.2 are more subtle than Sect. 5.1 not only because it is shown for grid-cell mean, but also because it is the result over the entire Northern Hemisphere. The positive and negative differences between the two versions in different areas partly offset each other. We have added '**over the NH**' in some sentences to make the descriptions more specific.

*Fig 9 - what is going on with the Tsoil between S0 and S1? For example, in panel e it looks like there is some sort of restart bug that prevents a smooth transition? Actually I am finding this whole figure confusing. What is grey vs. yellow? What is the x-axis? Is the S1 plotted in every figure, but just overplotted in some? Same questions about fig 10, with addition of wondering why the scale break for mean annual snow and why the values start so low?*

**[Response]** First, as shown in Table 1, S0 was run with MICT for the three periods (A, B, C), and S1 was run with MICT-teb for the three periods. For S1, the flags controlling the tilling energy budgets in MICT-teb were turned off during Periods A and B but were turned on during Period C. The non-smooth transition since Period C in S1 resulted from the start of tiling energy budgets in MICT-teb, not a restart bug. Before Period C, the results between S0 and S1 were identical because both were run without tiling energy budgets. To investigate the sole effects of tiling energy budgets on energy, water, and carbon in Sect. 5.1, it is necessary to provide the same starting point (and should also be an equilibrium). Otherwise, the differences between the two versions cannot be attributed solely to the tiling, but also to the different starting points.

Second, the grey, yellow, and light orange backgrounds indicate the three periods (A, B, C), respectively. We used the color backgrounds because it's hard to tell the length of each period only with the tick label on the x-axis.

Third, S1 is plotted in the figure, but it is covered by S2 during Period C. The energy budgets and water cycle can respond quickly when the tiling energy budgets were turned on during Period C in S1.

Fourth, Period A is a spin-up simulation where the vegetation starts to grow from a zero biomass and the energy budgets and the water cycle start from the initial values prescribed in the model. Some variables such as snow could be very low in the initial year.

To make Fig. 9 more easily understood, we have added more explanations in the figure caption: "**The backgrounds in Figs. (a), (c), (e), and (g) are colored to help explain the simulation length of the three periods. Detailed explanations for the three simulations (S0, S1, and S2) and the three periods (A, B, and C) can be found in Table 1 and Sect. 4.**".

*Fig 12 - Here it seems that SOC for all three simulations is higher than for the 'OBS' (side comment- this is not observed since it is a gridded product of a fundamentally point-scale phenomenon, perhaps 'observation-based' would be a more accurate name), but it looked from Fig S2 that MICT tends to not have the soil C be deep enough. Does that mean that the SOC is now deeper with MICT-teb or is it just higher amounts in shallower soils?*

**[Response]** First, we have used the word "**observation-based**" to name this gridded product as the reviewer suggested. Then for the total SOC for 0-3 m over the NH, all three simulations are higher than the observation-based product (the maximum depth is 3 m), suggesting that the model overestimates the SOC for 0-3 m (Fig. 12). This result cannot be compared with Fig. S2, because Fig. S2 only compares the vertical profile of SOC over 16 sites. The underestimation at site level could not be representative of the entire NH.

*Line 483 - same complaint as earlier about citing the proper refs for the model inputs.*
**[Response]** The references have been added in the revised version.

*Fig 14: The threshold fractions are the minimum fractional coverage of that land cover type in the gridcell? Unclear what is meant here... Also instead of 'SIM', please keep consistent and put what model was used to produce the values, i.e. MICT-teb.*

**[Response]** Yes, the threshold fractions are the minimum fractional coverage of that land cover type in the grid cell. We have added the explanation in the revised figure caption. The "SIM" has been replaced with "MICT-teb" in the revised figure as suggested.

*line 537: I am not really sure how Fig S20 relates to this discussion of a weak correlation between biases. Perhaps this could be made more clear?*

**[Response]** Thanks for the comment. Fig. S20 only shows the separate comparisons between simulated $T_{soil}$ or $T_{surf}$ and that from site observations or MODIS. The correlation between biases is computed independently, not from this figure. We have removed the figure to avoid confusion.

[revised manuscript text omitted]

---

## Author Comment (AC2)

**Response to the reviewers**

**General comments**

*Surface heterogeneity has large impacts on surface energy, water and carbon cycles. This study developed and evaluated a new multi-tile surface energy budget scheme in the ORCHIDEE-MICT model. Expectedly, the improved model indeed shows better performance in some regions, especially the permafrost areas. However, some issues are needed to be resolved first: 1) the research gap is not clear apart from using a different model; 2) many details on the methods are missing, e.g., the vegetation albedo calculation, the unreasonable assumption of 1 for the emissivity, the definition and estimations of surface temperature, the snow cover fraction parameterization, etc. 3) The statistical tests on the significance of the model-model and model-observation differences are needed; 4) The authors just compared the surface temperature between model and observations, and more comprehensive comparisons are needed using the benchmark datasets for all the surface energy balance variables, also the carbon and water cycles-related variables. Please see below for my specific comments.*

**[Response]** Thank you so much for your time in reviewing our manuscript and for providing your constructive comments and suggestions. In response to the reviewer's comments, we 1) reorganized the introduction and discussion to provide a complete description of the research gap and tiling in different land surface models; 2) provided additional details in the methods section, explaining the calculation of albedo, the value of emissivity, the definition of surface temperature, and the calculation of snow cover fraction; 3) explained the calculation and applicability of significance tests for the model-model and model observation differences in this study; and 4) added the comparison of other components of the surface energy budget including albedo and latent flux between the simulations and the MODIS products. Please find detailed point-by-point responses to each major concern and small comment below.

**Major concerns**

*1. In the abstract, I suggest the authors provide some quantitative metrics for the performance of the improved and original models.*

**[Response]** As the reviewer suggested, we added quantitative metrics into the revised abstract (copied as below, L23-30).

"**With the specific values of surface properties for each vegetation type, the new version presents warmer surface and soil temperatures (~0.5 °C, 3%), wetter soil moisture (~10 kg m$^{-2}$, 2%), and increased soil organic carbon storage (~170 PgC, 9%) across the Northern Hemisphere. … However, the separation of sub-grid energy budgets in the new version improves permafrost simulation greatly by accounting for the presence of discontinuous permafrost types (~ 3 million km$^2$) …**"

*2. In the second paragraph of the introduction section, the authors only used the surface temperature as one example to introduce the background. However, surface temperature is just one import factor in the surface energy budgets. Actually, there are already many existing studies analyzing the impacts of surface heterogeneity on surface energy balance and water cycles, as well as land-atmosphere interaction. I suggest the author reorganize this paragraph to better introduce the existing studies and background.*

**[Response]** Thanks for this comment. We have reorganized the logic of this paragraph and have included more existing studies analyzing the impacts of surface heterogeneity on other components of surface energy balance besides the surface temperature, copied here as below (L41-77).

[revised manuscript text omitted]

*3. Considering that the multi-tiling scheme has been used in other land surface models, please clarify the research gaps apart from the specific model used in the study.*

[Response] Following our last response to the reviewer's comment 2, we added more text about the background why we need to represent the tiling energy budgets. To connect with ORCHIDEE-MICT, we have added one sentence in the revised introduction "**To enable the representation of surface heterogeneity and open the door to a series of new landforms and processes,** we implement tiling energy budgets at the surface and subsurface for each plant function type (PFT) in a state-of-the-art LSM, ORCHIDEE-MICT …" (L79-83). We have also added the section 6.3 in the discussion, covering the reason why we performed the improved tiling development in ORCHIDEE-MICT, the extra computation cost of the new version, and the reminder on when and where to tile for the potential users of our version (copied as below, L639-649).

"**6.3 Remarks on the tiling land surface scheme**
**The tiling work in this study was initiated because of the planned introduction of new arctic landforms for permafrost regions into ORCHIDEE-MICT that requires independent energy**

**budgets, carbon and water cycles. The decision to tile by PFT, rather than other units, was determined based on the current model structure. In the new version, additional variables with a new PFT dimension were introduced only for the energy module, rather than all modules, resulting in a 15-20% slower run time compared to the initial version. Recently, several other model groups have also been working on the tiling and the evaluation of its impacts on existing and new processes, such as JULES (Rumbold et al., 2023) and CLASS (Melton et al., 2017). The implementation of tiling in different land surface models can be compared to inspire other groups planning to represent sub-grid heterogeneity of energy budgets in their models. Moreover, for potential users of our new version, it is worth reminding them to carefully consider when and where to apply tiling in their studies to optimize research objectives and computational costs.**"

*4. In section 2, Figure 1: It is unclear how many soil/snow columns are included for each grid cell for the improved and original versions. Whether do different PFTs have different snow cover and soil characteristics or not? These (especially the snow cover) may have big impacts on surface energy balance. Besides, the authors used the standard rectangle grids to represent different PFTs, which may mislead the authors, because the same PFTs may distribute in different sub-grids.*

**[Response]** In the improved and original version, every PFT has its own snow and soil layers, but the original version used the grid-cell mean energy budgets for all PFTs, and thus the same snow cover fraction for each PFT. As pointed out by the reviewer, the snow cover, as well as many other energy variables are different among different PFTs, especially grasses, shrubs and trees, which is one main reason why we started the tiling work for energy. Now in the improved version, the energy budgets of surface, snow and soil are PFT-specific. Regarding Fig. 1, the big rectangle indicates a grid cell in the model, with each sub-rectangle for one PFT. There are 16 PFTs at maximum (set in our simulation and can be changed in other simulations) existing in one grid cell and one PFT can only be distributed in its sub-rectangle in ORCHIDEE-MICT. Together with the comment from the other reviewer, we made some modifications to this figure (copied here as Figure R1), and we also added some key words in the figure caption to avoid misleading interpretation.

[Figure]

Figure R1 (Figure 1). Schematic representation of energy budgets at the surface, snow layers, and soil layers **in one grid cell** of ORCHIDEE-MICT (MICT) (a) and the new tiling energy budget version (MICT-teb) (b). $SW_{in}$, $SW_{out}$, $LW_{in}$, $LW_{out}$, H, and, LE represent incoming ShortWave radiation, outward ShortWave radiation, incoming LongWave radiation, outward LongWave radiation, sensible heat flux, latent heat flux, respectively. PFT indicates Plant Function Type. There are 3 layers for snow, and 32 layers for soil **for each PFT** in the model. In MICT, $SW_{in}$, $SW_{out}$, $LW_{in}$, $LW_{out}$, H, and heat fluxes in snow and soil layers are calculated as grid-cell mean but LE is calculated for each PFT, while in MICT-teb, all of the heat fluxes are calculated for each PFT. **The red and blue arrows distinguish the grid-cell mean and PFT-specific calculation.**

*5. In equation 2, the emissivity is assumed to 1, which is not reasonable. For vegetation, the emissivity depends on LAI. Different land types also show very different emissivity.*

**[Response]** The value of 1 for the emissivity is used as default by ORCHIDEE. It is defined as a coefficient describing the capacity of a body to emit radiation. According to the classical geography textbook *Principles of Terrestrial Ecosystem Ecology (2ⁿᵈ Edition)* by F. Stuart Chapin, III, the emissivity is about 0.98 in vegetated ecosystems (page 96). And according to the textbook *Climate Change and Terrestrial Ecosystem Modeling* by Gordon Bonan, most objects have a broadband emissivity of 0.95–0.98 when integrated over all wavelengths (page 42). Thus, using the value of 1 for the emissivity could be rough, but not too unreasonable.

*6. In equation 2, I am also confused about how did the authors define the surface temperature here, because the surface temperature can change within the vegetation canopy and understory background. Please clarify how the model calculated surface temperature here as well as the relationship between surface temperature, canopy temperature and ground temperature.*

**[Response]** Since ORCHIDEE-MICT is not capable of calculating the leaf energy budgets, there's only one surface temperature (i.e., without height dimension) in the model. This surface temperature is used to calculate all surface energy fluxes. We added one sentence to avoid confusion in the revised version:

**"Since MICT is not capable of calculating leaves energy budgets separately from soils and stems, there's no vertical-layered temperature from the ground surface to the top of the canopy, and thus the $T_{surf}$ here is used to calculate all surface energy fluxes."** (L138-140).

*7. Line 117: How did the authors retrieve the soil albedo for the vegetated regions from remote sensing data?*

**[Response]** The bare soil albedo used in the model is not directly retrieved from remote sensing data. The solar radiation absorbed by the vegetation layer and the background is separated using the Joint Research Centre Two-stream Inversion Package (JRC-TIP). ORCHIDEE uses the background albedo as bare soil albedo. More details about the package, JRC-TIP can be found in Pinty et al., (2011). We have added the reference in the revised manuscript.

*8. Section 2.3: How did the authors set the empirical values for different snow-related parameters?*

**[Response]** These empirical values come from the PhD thesis of Sophie Najm Chalita (1992) and the corresponding journal article (Chalita et al., 1994), where the parameterization of the snow albedo has been evaluated with site measurements from Robinson and Kukla, (1984). We added the reference in the revised version.

*9. Section 2.1: It is unclear how the model calculates the surface albedo.*

**[Response]** In ORCHIDEE-MICT, the grid-cell surface albedo is calculated as the area-weighted average of the albedo across all PFTs in this grid cell. When snow is free, the area-weighted average calculation only considers the albedo of bare soil (prescribed with remote sensing data) and vegetated regions (prescribed with constant values, Table S1). When snow is covered, the albedos of snow-covered bare soil and the vegetated area use the prescribed values as listed in Table S3. We have revised the description of the calculation of surface albedo following the last two comments of the reviewer.

*10. what is the reference of this equation? Are the impacts of topography on snow cover fraction considered in this equation?*

**[Response]** The references for this equation (Niu and Yang, 2007 and Wang et al., 2015) have been added to the revised manuscript. The impacts of topography on snow cover fraction have not been considered in this equation yet.

*11. Section 3.3: There are already high resolution (e.g., 250m) SOC datasets at the global scale, e.g., soilgridv2.*

**[Response]** Thanks for this comment. The soilgrid v2 does have a high resolution, but it only provides the SOC for 0-2 m. We cannot use it because the maximum depth of the soil layer is as deep as 38 m in

the model and the maximum depth for SOC in most permafrost regions, especially for peatlands, can be as deep as 10 m.

12. *Section 4: I am curious about why did the authors run the simulations cycling the forcing from 1901-1920 rather than the present-day simulations? The present-day simulations can be more suitable for the comparisons with the available remote sensing data.*

**[Response]** We ran both simulations with the forcing cycling 1901-1920 and with the present-day forcing in the model in fact, but the present-day simulation was not mentioned in the original Section 4, but was put in Section 6 when we were comparing the improved version with the remote sensing data. As described in Section 4, the simulations during periods A and B were run to accumulate a SOC stock under the same conditions as observed in reality while the simulation during period C was run to achieve near-equilibrium carbon and water cycles. The three periods are indispensable before running a present-day simulation. If we only aim to compare the muti-tiling energy budgets from the improved version and the single-tiling energy budgets from the original version, it is enough to run the simulations by period C. But if we want to compare the improved version with remote sensing data, we need to extend the simulation to the present day. To make the description clearer, we have rearranged the text (copied as below, L281-286) and Table 1, to add period D (Transient simulation) in Section 4.

"All of the three groups are run for the NH (0°-90°N) at a spatial resolution of 2° × 2°, with **four** simulation periods: A) Spin-up1, 100 years of the full ORCHIDEE with a looped 1901-1920 climate, $CO_2$ level of 1901 at 296.80 ppm, and the land cover map of 1901; B) SubC, 10,000 years of the soil carbon sub-model (offline) to accumulate SOC; C) Spin-up2, 50 years of the full ORCHIDEE to reach equilibrium with a looped 1901-1920 climate, $CO_2$ level of 1901, and the land cover map of 1901; **and D) Transient simulation, the full ORCHIDEE with varying climate, $CO_2$ level and land cover maps from 1901 to 2020.**"

13. *Figure 4-5 and 7-8: Please show whether the differences are significant or not for each grid cell. Table 2: Please also clearly define the magnitude of one arrow and two arrows. The statistical tests on the case differences are also needed in section 5.2 for all variables.*

**[Response]** Thanks for the comment. In Figures 4-5 and 7-8, the differences between MICT-teb and MICT were calculated using the 10-year mean results from period C. That is, there's only one value for each model. Thus, the significance testing is not applicable for these figures. For Table 2, it is a qualitative summary of the change in surface temperature and the changes in surface properties. We provided this summary, hoping it helps readers understand the modifications done in the new version i.e., the changes in surface properties and the consequences after modifications i.e., the change in surface temperature. The significance testing is not applicable, either, but the quantitative difference in the variables in Table 2 can be found in Figs. 4-5 for grid scale and in Figs. 9-11 for the whole NH.

*14. Section 6.1: When comparing with MODIS data, did the authors extract the model values in the MODIS overpass time? Besides, also show whether the differences between model estimates and MODIS are significant or not. Did the improved version show better performance than the original version?*

**[Response]** We only compared the average from 2001 to 2020 between the model and MODIS data, so significance testing is not applicable. Including the representation of PFT-specific energy budgets alleviates the LST bias from MICT in some areas such as western North America and northern Europe and in some seasons such as autumn in high latitudes, but at the same time, it aggravates the LST bias in some areas and some seasons such as all seasons in tropical regions (Fig. 13(f)).

*15. The authors just compare the surface temperature. I suggest the authors compare all the surface energy balance components with the available remote sensing data or reanalysis data, e.g., surface albedo, net radiation, etc. Comparing all of them can give more hints on the model improvement and potential drawbacks.*

**[Response]** Thanks for the constructive comment. Since the comparison of surface temperature has mirrored the comparisons of outward longwave radiation (LWout) and sensible flux (H), we further compared the albedo (mirroring net shortwave radiation) and the latent heat flux (LE) between the model and MODIS data, including the spatial patterns and the seasonality, to give a complete evaluation of the simulated surface energy balance as the reviewer suggested. The revised text and figures were copied as below (L500-552):

[revised manuscript text omitted]

To sum up, the two model versions can reproduce the spatial patterns of the three components of surface energy budgets from MODIS basically, but with biases in some areas and some seasons due to the under-representation of some processes in the model and / or the uncertainties in satellite observation. Including the PFT-specific energy budgets can alleviate the bias in some areas and some seasons, but at the same time aggregates the biases in other areas and seasons, which suggests the uncertainties from the model cannot be solved by the representation of PFT-specific energy budgets but relies on the incorporation / refinement of some key processes.

**Minor concerns**

*1. Line 53: In some ESMs, e.g., E3SM, the topography-based tiling scheme has also been used.*
**[Response]** Thanks. We have added it in the revised version (L54-55), as an example how ESMs consider the topography into the tiling scheme.

*2. Line 100: Modify the captions of Figure 1.*
**[Response]** It has been corrected.

*3. Line 264: what is the meaning of "offline" here?*

**[Response]** It means that the soil carbon module is run independently, not coupled with the hydrology and energy. We have removed it to avoid the confusion with the "offline" used for land surface models.

*4. Line 290: why did the author select the three grids?*

**[Response]** They were selected randomly, only to show the results for the three latitudes. We have added "**randomly**" in this sentence.

*5. Line 306: Looks like such calculation is not accurate because of neglecting the light interaction between canopy and soil.*

**[Response]** As mentioned earlier, the canopy energy budgets are absent in current ORCHIDEE-MICT. It is under developing for the ORCHIDEE model (Alléon et al., 2023) and has not been included in ORCHIDEE-MICT. This could contribute to the disagreements between the simulations and the satellite datasets in Section 6.1.

[revised manuscript text omitted]